# Rethinking Evaluation Paradigms in IBP-based Certified Training

**Konstantin Kaulen** [1]   **Hadar Shavit** [1]   **Holger H. Hoos** [1 2]

## Abstract

Deep neural networks achieve strong performance on many supervised learning tasks but remain vulnerable to adversarial perturbations. Neural network verification provides mathematically rigorous robustness guarantees, yet at substantial computational cost. To mitigate this, certified training techniques optimise for verifiable robustness during training, typically inducing a trade-off between *natural* and *certified accuracy* controlled by method-specific hyperparameters. Because these metrics are inherently conflicting, the common practice of reporting a single configuration is problematic: it can mislead conclusions about overall performance and prevents unbiased assessments of the state of the art. We address this by evaluating certified training methods via *Pareto front* comparisons over the natural–certified accuracy trade-off. To enable fair, method-agnostic comparisons, we perform efficient automated multi-objective hyperparameter optimisation to identify a set of Pareto-optimal configurations for each method. This approach often uncovers substantial undertuning in previously reported configurations, yielding superior performance and establishing a new state of the art. Leveraging these fronts, we present the first comprehensive multi-objective comparison of certified training approaches, showing that prior advancements are less pronounced than assumed and revealing previously unreported performance complementarities.

## 1. Introduction

In recent years, deep learning has enabled remarkable advances across several application areas ranging from computer vision (Dosovitskiy et al., 2021) to protein structure

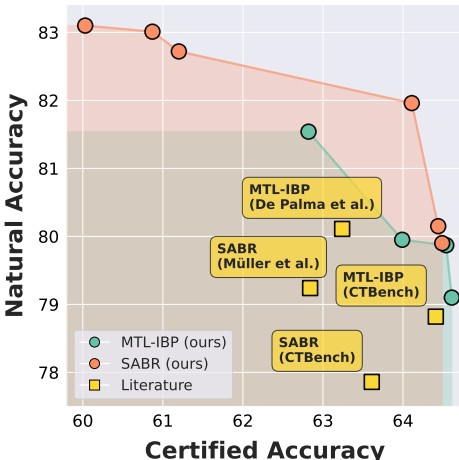

*Figure 1.* Pareto fronts of SABR (Müller et al., 2023) and MTL-IBP (De Palma et al., 2024b) on CIFAR-10 with $\epsilon = \frac{2}{255}$, obtained using our multi-objective evaluation, compared to results from the original publications and CTBench (Mao et al., 2025). By moving beyond single-objective tuning, our approach identifies Pareto fronts that dominate previously reported results and exposes how conclusions drawn from single configurations and non-standardised tuning can misrepresent the true state of the art.

prediction (Jumper et al., 2021). Concurrently, there has been a fast-growing trend towards employing deep-learning-based systems in safety-critical domains, such as unmanned aircraft manoeuvre advisory systems (Julian et al., 2019). However, it is well known that deep neural networks are vulnerable to *adversarial examples* (Szegedy et al., 2014): inputs perturbed by small, carefully designed modifications that lead to misclassification (see, *e.g.*, Goodfellow et al., 2015; Madry et al., 2018).

While adversarial attacks are widely used to expose vulnerabilities in neural networks, their heuristic nature can prevent them from discovering adversarial manipulations, even when these exist. Thus, *neural network verification* techniques have been proposed that provide *formal guarantees* on the robustness of neural networks (see, *e.g.*, Tjeng et al., 2019; Wang et al., 2021; Ferrari et al., 2022; De Palma et al., 2024a). These come at the cost of substantially increased computational requirements, since proving even simple properties is an $\mathcal{NP}$-complete task (Katz et al., 2017; Sälzer & Lange, 2021).

[1]Chair of AI Methodology, RWTH Aachen University, Aachen, Germany [2]LIACS, Leiden University, Leiden, The Netherlands. Correspondence to: Konstantin Kaulen <kaulen@aim.rwth-aachen.de>.

*Proceedings of the $43^{rd}$ International Conference on Machine Learning*, Seoul, South Korea. PMLR 306, 2026. Copyright 2026 by the author(s).

To mitigate this computational burden, *certified training* techniques can be employed to produce networks for which formal guarantees can be obtained more efficiently (see, *e.g.*, Gowal et al., 2019; Zhang et al., 2020; Shi et al., 2021; Müller et al., 2023; Mao et al., 2023; De Palma et al., 2024b). Here, *incomplete verification* methods that yield sound, but potentially loose, bounds on the outputs of the neural network are employed to over-approximate the worst-case loss within a given threat model. Currently, state-of-the-art methods rely on *interval bound propagation* (IBP) (Gowal et al., 2019) for the bounding process.

While the resulting networks are more amenable to formal verification techniques, they generally perform far worse on clean data in comparison to conventional neural network training (see, *e.g.*, Müller et al., 2023; De Palma et al., 2024b). This effect is known as the *robustness-accuracy trade-off* in the context of adversarial training (see, *e.g.*, Tsipras et al., 2019; Zhang et al., 2019), but remains mostly unexplored for deterministic certified training methods. However, understanding and improving on this trade-off is crucial for making certified training methods applicable in practice.

To govern this trade-off between robustness and accuracy, certified training techniques expose hyperparameters. In particular, they introduce a weighting factor to balance the certified loss obtained through IBP against either clean loss (Gowal et al., 2019; Zhang et al., 2020) or adversarial loss (Müller et al., 2023; De Palma et al., 2024b). Until now, the state of the art in certified training has been determined by tuning methods to one specific trade-off that improves over results from related work; manually (see, *e.g.*, Müller et al., 2023; De Palma et al., 2024b) or by relying on grid search (Mao et al., 2025).

However, due to the robustness-accuracy trade-off, the problem gives rise to a *Pareto front* of configurations, *i.e.*, a set of configurations for which improving one objective necessarily degrades the other. To date, this front has not been systematically explored in the context of certified training. Previous works compared only single configurations, rendering comparisons highly dependent on the chosen point in the trade-off space and thus potentially biased.

In this work, we argue for a paradigm change regarding the assessment of the state of the art in certified training. Contrary to the currently prevailing evaluation methodology, we advocate for a comparison of certified training techniques using their Pareto fronts over the complete trade-off space. To this end, we propose a simple and efficient method to discover Pareto sets of well-performing configurations using automated multi-objective hyperparameter optimisation. To render this approach tractable in practice, we rely on several carefully designed adjustments to the standard optimisation algorithm, such as using certified robustness derived from cheap incomplete verification as a proxy objective for the final robustness objective assessed using costly complete verification. These adjustments enable the discovery of a Pareto front using evaluation budgets similar to those required in related work for identifying individual configurations. By fixing the evaluation budget and using a general, expert-designed search space, we conduct a principled and fair comparison of certified training methods across the complete trade-off space. In addition, we find that most certified training methods yield previously unreported Pareto-optimal trade-offs that dominate prior results, highlighting evaluation inconsistencies and possible under-tuning in the existing literature. In Figure 1, we highlight how single-objective tuning can obscure the true performance of certified training methods, while Pareto-front evaluation enables a fair and informative comparison.

By aggregating the Pareto fronts of all methods into a single joint front per benchmark, we reassess the state of the art across the entire trade-off space and show that claimed advancements of recent years are far less pronounced than suggested in prior work, especially for larger perturbation radii. Furthermore, we show that there is no single certified training method that uniformly achieves state of the art performance; instead, methods are complementary, with SABR (Müller et al., 2023) generally excelling at high natural accuracy and MTL-IBP (De Palma et al., 2024b) delivering stronger certified guarantees. Importantly, we show that the trade-off is governed by complex interactions between multiple hyperparameters that could only be uncovered using state-of-the-art multi-objective Bayesian hyperparameter optimisation methods. By systematically discovering the full Pareto fronts, we provide the first multi-objective perspective on certified training, establish a new evaluation standard for the community and offer actionable insights for designing, tuning and comparing methods across the complete robustness–accuracy spectrum.

## 2. Background

In the following, we provide the necessary background, covering neural network verification and certified training.

### 2.1. Neural Network Verification

Generally, given a neural network $f_\theta : \mathbb{R}^d \mapsto \mathbb{R}^c$, $c, d \in \mathbb{N}$ that maps inputs $\mathbf{x} \in \mathbb{R}^d$ to outputs $f_\theta(\mathbf{x}) \in \mathbb{R}^c$, *formal neural network verification* is concerned with proving whether a given input-output property *holds* or *is violated* for $f$.

In this study, we focus on classification problems with scalar labels $y \in \mathbb{N}$ and on *local robustness* in an $\ell_\infty$ norm ball with radius $\epsilon$ denoted as $\mathcal{B}_\infty^\epsilon$. Given an input $\mathbf{x}_0$ with label $y_0$ and the norm ball $\mathcal{B}_\infty^\epsilon = \{\mathbf{x} \mid \|\mathbf{x} - \mathbf{x}_0\|_\infty \leq \epsilon\}$, the

local robustness problem can be stated as

$$\forall \mathbf{x}' \in \mathcal{B}_{\infty}^{\epsilon} : \arg\max_{j} f_{\theta}(\mathbf{x}')[j] = y_0 \qquad (1)$$

, where $\mathbf{v}[i]$ denotes the $i$-th entry of vector $\mathbf{v}$. The problem reduces to computing the sign of the following optimisation problem, where $\mathbf{z}(\mathbf{x}, y) \in \mathcal{R}^c$ is defined as the vector of logit differences, *i.e.*, $\mathbf{z}(\mathbf{x}, y) := f_{\theta}(\mathbf{x})[y] \cdot \mathbf{1} - f_{\theta}(\mathbf{x})$:

$$\min_{\mathbf{x}' \in \mathcal{B}_{\infty}^{\epsilon}} \min_{i \neq y} \mathbf{z}(\mathbf{x}', y)[i] \qquad (2)$$

Computing an exact solution to Equation 2 is known to be an $\mathcal{NP}$-complete problem (Katz et al., 2017; Sälzer & Lange, 2021). Therefore, in practice, sound lower bounds to the logit margins $\underline{\mathbf{z}}(\mathbf{x}, y)[i] \leq \mathbf{z}(\mathbf{x}, y)[i], i \in \{1, \ldots, c\}$ are approximated using *incomplete* verification methods (see, *e.g.*, Zhang et al., 2018; Singh et al., 2019; Xu et al., 2021) that can be used within the *branch and bound* framework (Bunel et al., 2020) to achieve completeness (see, *e.g.*, Wang et al., 2021; Ferrari et al., 2022; De Palma et al., 2024a). Despite considerable progress in improving the efficiency of complete verification (see, *e.g.*, König et al., 2022; Zhou et al., 2024; 2025; Kaulen et al., 2025a; Duong et al., 2025), solving challenging properties within reasonable compute budgets often remains infeasible (Kaulen et al., 2025b).

### 2.2. Training Robust Neural Networks

Madry et al. (2018) introduced the problem of training robust neural networks as a min-max optimisation problem that aims to find parameters $\theta$ that minimise an expected worst-case loss measured through $\mathcal{L} : \mathbb{R}^c \times \mathbb{N} \to \mathbb{R}$ in the $\ell_{\infty}$ norm ball around samples from a data distribution $(\mathbf{x}, y) \sim D$:

$$\theta \in \arg\min_{\theta'} \mathbb{E}_D \left[ \max_{\mathbf{x}' \in \mathcal{B}_{\infty}^{\epsilon}} \mathcal{L}(f_{\theta'}(\mathbf{x}'), y) \right] \qquad (3)$$

As mentioned previously, calculating the exact worst-case loss is computationally not feasible, since it is equivalent to solving Equation 2. Therefore, Madry et al. (2018) under-approximate the inner maximisation by means of *projected gradient descent* (PGD), which iteratively searches for points $\mathbf{x}_{\text{adv}}$ in $\mathcal{B}_{\infty}^{\epsilon}$ that maximise the worst-case loss. We refer to this as the *adversarial loss* $\mathcal{L}_{\text{adv}} := \mathcal{L}(f_{\theta}(\mathbf{x}_{\text{adv}}), y)$. While the resulting networks are empirically robust, *i.e.*, far more resistant to adversarial attacks than traditionally trained networks, they do not yield certifiable guarantees and may be vulnerable to stronger adversarial attacks (Mao et al., 2025; Croce et al., 2021). *Certified training* methods follow an orthogonal approach by over-approximating the true value of the inner maximisation by means of incomplete verification methods. The *verified loss* $\mathcal{L}_{\text{ver}}$ is computed on the previously defined lower bound to the logit differences of $f_{\theta}$ (Wong & Kolter, 2018):

$$\max_{\mathbf{x}' \in \mathcal{B}_{\infty}^{\epsilon}} \mathcal{L}(f_{\theta}(\mathbf{x}'), y) \leq \mathcal{L}_{\text{ver}} := \mathcal{L}(-\underline{\mathbf{z}}(\mathbf{x}, y), y) \qquad (4)$$

This loss decreases when the employed incomplete verifier can prove that $f_{\theta}$ is locally robust for the given training sample. Perhaps surprisingly, training methods that employ a hyper-box relaxation via *interval bound propagation* (IBP) (Gowal et al., 2019) to bound the outputs of the network currently yield best results, despite relying on a relatively loose over-approximation (see, *e.g.*, De Palma et al., 2024b; Mao et al., 2024; Müller et al., 2023). We present the concrete certified training approaches relevant to this work in Section 3 and give a thorough explanation of IBP in Appendix B.1. Generally, certified training methods are evaluated with regard to two metrics, *i.e.*, *clean* and *certified* accuracy, where, given a test set, the former refers to the fraction of correctly classified inputs and the latter refers to the fraction of inputs for which the network is provably robust within $\mathcal{B}_{\infty}^{\epsilon}$ assessed using complete verification.

## 3. Related Work

In the following, we give a brief overview of the state of the art in certified training and of the prevailing evaluation methods in the field.

**State-of-the-Art Certified Training Techniques.** As stated previously, state-of-the-art certified training relies on IBP to approximate the worst-case robust loss. This approach was first introduced by Gowal et al. (2019) but required gradually increasing $\epsilon$ to its final value over hundreds of *ramp-up* epochs to stabilise training. In addition, Gowal et al. introduced a trade-off parameter $\kappa$ that is decreased from 1 to 0 during ramp-up, weighing clean and verified loss: $\kappa \cdot \mathcal{L}(f_{\theta}(\mathbf{x}), y) + (1 - \kappa) \cdot \mathcal{L}_{\text{ver}}(f_{\theta}(\mathbf{x}), y)$. Prior to certified training, the network may be initialised with several *warm-up* epochs using the clean loss. Zhang et al. (2020) propose to combine IBP and CROWN (Zhang et al., 2018) bounds in *CROWN-IBP* to compute $L_{\text{ver}}$. Here, CROWN relaxations are used to bound the final output based on IBP bounds of intermediate layers. Furthermore, a transition is made from CROWN-IBP to IBP bounds during ramp-up, using an additional trade-off parameter $\beta$. Xu et al. (2020) further reduce the complexity of CROWN-IBP through *loss fusion*, a technique that enables direct computation of the verified loss without requiring logit differences. Shi et al. (2021) suggest the use of BatchNorm layers (Ioffe & Szegedy, 2015) and introduce specialised initialisation and regularisation techniques resulting in shorter ramp-up schedules and better performance. More recently, a line of research emerged that combines certified and adversarial losses. Müller et al. (2023) compute an unsound verified loss called *SABR* by propagating a smaller subset of the input region with edge length $\tau \cdot \epsilon$ using IBP. The centres of the hyper-box are identified using PGD. Additionally, *ReLU shrinking* is used to reduce the magnitude of IBP bounds by multiplying them with a constant $c < 1$ before each activa-

tion, thereby gradually increasing focus on adversarial loss. De Palma et al. (2024b) show that loss functions conceptually similar to SABR can be obtained by considering convex combinations of $\mathcal{L}_{ver}$ and $\mathcal{L}_{adv}$ weighed by $\alpha$. Among those, the *MTL-IBP* loss is defined as $\alpha \cdot \mathcal{L}_{ver} + (1 - \alpha) \cdot \mathcal{L}_{adv}$. An effect similar to ReLU shrinking is achieved by carrying out adversarial attacks over a larger perturbation radius.

**Evaluation of Certified Training.** To assess certified accuracy of trained models, related work employed state-of-the-art complete verification systems *Oval* (De Palma et al., 2024a) or *MN-BaB* (Ferrari et al., 2022). In addition, the tuning of parameters including the learning rate, the number of warm- and ramp-up epochs and trade-off parameters, such as $\kappa$ or $\alpha$, is crucial for achieving state-of-the-art performance. Until now, researchers have mostly relied on tuning parameters manually to obtain a single configuration that compares favourably to the current state of the art (see, *e.g.*, Shi et al., 2021; Zhang et al., 2020; Müller et al., 2023; De Palma et al., 2024b; Mao et al., 2023). Recently, Mao et al. (2025) proposed *CTBench*, a novel benchmark for certified training, with the goal of ensuring a fair comparison between methods by employing grid search over separately designed hyperparameter spaces per benchmark, which required up to 250 evaluations. While the CTBench benchmark uncovered that certified training techniques can achieve stronger certifiable guarantees when they are systematically tuned and consistently implemented, the results presented in their work were obtained by tuning to one specific trade-off that often favoured certified accuracy and, thus, came at the expense of markedly reduced clean accuracy on some benchmarks as we show in Section 5.

## 4. Pareto-Based Evaluation of Certified Training Methods

As mentioned previously, the performance of certified training techniques along the trade-off between natural and certified accuracy can be controlled by tuning the hyperparameters of the respective methods. However, to date, the hyperparameter optimisation problem has been treated as a single-objective problem with optimising for certified accuracy only (Mao et al., 2025) or for a sum of clean and certified accuracy (De Palma et al., 2024b), with the majority of publications not disclosing the objective and procedure of their tuning process (see, *e.g.*, Müller et al., 2023; Mao et al., 2023; Shi et al., 2021; Zhang et al., 2020). This highlights that there exists no clear view on what defines a well-performing hyperparameter configuration with regard to the robustness-accuracy trade-off in the context of certified training. In addition, there is a lack of an unbiased, fair and standardised evaluation procedure that can be easily employed to position new methods in comparison to the current state of the art.

To harmonise the differing views in evaluating IBP-based certified training, we argue for a new evaluation paradigm. Instead of assuming a definitive answer to whether natural or certified accuracy should be prioritised, we advocate to compare methods over the entire trade-off space. In doing so, we offer a nuanced perspective on performance that enables researchers and practitioners to select the most suitable method based on their requirements – whether they prioritise strong natural accuracy, strong certified robustness, or any trade-off between the two.

In the following, we present a simple and efficient method for the discovery of a Pareto-optimal set of hyperparameter configurations for certified training based on the well-known concept of multi-objective hyperparameter optimisation.

**Search strategy.** Since hyperparameters are often interdependent (Moosbauer et al., 2021), we jointly optimise all hyperparameters within a method-specific search space. In addition, we desire to exclude uninteresting regions of the Pareto front from the search, *i.e.*, configurations exhibiting high natural accuracy with extremely low certified accuracy, or *vice versa*, which can be obtained, *e.g.*, by tuning SABR and MTL-IBP to reduce to adversarial training. Therefore, our optimisation algorithm needs to be constrained to an area of interest to avoid spending expensive resources on uninteresting configurations.

Due to these reasons, we employ multi-objective Bayesian optimisation with a Gaussian process surrogate and an EHVI acquisition function that is capable of accommodating constraints (Daulton et al., 2020) (see Appendix B.2).

Since the optimisation objectives are independent from each other, we model them using distinct Gaussian processes. To avoid undesirable outcomes, such as becoming trapped in local optima or over-exploration of specific parts of the Pareto front, we execute the optimisation with three pseudo-random seeds. We then combine the Pareto fronts discovered by those three runs to create one single Pareto front.

**Search space design.** Since the influence of hyperparameters on performance is not known *a priori*, we opted to include all relevant hyperparameters in our search space. These include general hyperparameters of deep learning pipelines, such as the learning rate, epochs at which the learning rate is decayed and the optimiser used to find best performing parameters with regard to Equation 3 (*e.g.*, Adam (Kingma & Ba, 2015) or RAdam (Liu et al., 2020)). Furthermore, we include $\ell_1$ regularisation, since it has proven beneficial for certified training, and optimise its weight-parameter. Regarding techniques specific to certified training, we optimise for the weight of the regularisation proposed by Shi et al. (2021), which is employed in all state-of-the-art methods (see, *e.g.*, De Palma et al., 2024b;

Müller et al., 2023). Furthermore, we search for an optimal number of warm-up and ramp-up epochs. It may also be beneficial to train with a larger perturbation radius than used for evaluation (see, *e.g.*, De Palma et al., 2024b; Gowal et al., 2019); hence, we optimise a parameter that scales the $\epsilon$ value used in training. Moreover, we search for optimal method-specific trade-off parameters $\tau$ for SABR- and $\alpha$ for MTL-IBP-based training. Regarding $\kappa$, we optimise two parameters $\kappa_{\text{start}} \geq \kappa_{\text{end}}$ and transition from $\kappa_{\text{start}}$ to $\kappa_{\text{end}}$ during the ramp-up phase. We handle the $\beta$ parameter in CROWN-IBP analogously. For SABR and MTL-IBP, we additionally optimise the number of PGD steps and their step size. To keep training cost tractable, we do not allow multiple restarts of the PGD attack, as done by Mao et al. (2025); a choice consistent with several prior studies (see, *e.g.*, De Palma et al., 2024b; Madry et al., 2018). Lastly, we tune the $\epsilon$-radius over which the PGD attack is carried out.

Overall, we constructed the search space to include all plausible parameter choices, rather than restricting it to those previously shown to be successful in the literature. If those choices were indeed optimal, we rely on the optimiser to discover them during search. For example, we included $\kappa$ and $\beta$ as optimisable parameters, while related work has deemed those transitions unnecessary, and we allow up to five warm-up epochs, while related work employed at most one (see, *e.g.*, Mao et al., 2025; Shi et al., 2021). With this, we aimed for an unbiased evaluation that may uncover previously unexplored configurations. We present the complete search space in Appendix E.7.

**Optimisation metrics.** While evaluating clean accuracy is cheap, evaluating certified accuracy with complete verification systems for each configuration is computationally infeasible. Thus, we optimise for an under-approximation of the true certified accuracy by employing the incomplete verification methods IBP, CROWN-IBP and CROWN, running computationally more demanding methods only when cheaper methods could not provide a result.

**Complete verification.** To obtain the final Pareto front, we assess the performance of all Pareto-optimal configurations found with regard to incomplete verification using a state-of-the-art complete verification system. However, the front may include several configurations with negligible performance differences, for which complete verification would incur unnecessary costs. Therefore, in cases where more than 5 configurations are part of the Pareto front, we employ single-linkage clustering (Sibson, 1973), which starts by assigning each configuration to its own cluster and then iteratively merges close clusters whenever the Euclidean distance between the metrics of configurations from two clusters is less than $d_{\text{min}}$. We evaluate one random configuration for each cluster and construct the final Pareto front

using the certified accuracies obtained through complete verification.

# 5. Empirical Evaluation

In the following, we present an evaluation of the current state of the art in IBP-based certified training using our novel multi-objective evaluation procedure.

**Setup of experiments.** We implemented our novel evaluation method in the open-source certified training library CTRAIN (Kaulen & Hoos, 2025).[1] As the methods under investigation, we focused on IBP, CROWN-IBP, SABR and MTL-IBP. In Appendix D.4, we provide additional results for the CC-IBP and Exp-IBP losses, which constitute convex combinations of the adversarial and certified loss conceptually similar to MTL-IBP (De Palma et al., 2024b). With this, we aimed to include methods that have credible claim to be state of the art as well as seminal advancements from the field. For the hyperparameter optimiser, we used BoTorch (Balandat et al., 2020) within the Optuna package (Akiba et al., 2019), which provides an implementation of the chosen optimisation algorithm. Based on preliminary experimentation, we set the evaluation budget for each optimisation run to 100 trials, resulting in 300 trials per benchmark. For complete verification, we used the state-of-the-art (Kaulen et al., 2025b) verification system $\alpha\beta$-CROWN (Wang et al., 2021; Xu et al., 2021) with a cutoff of $1\,000$ seconds in wall-clock time. For comparability with related work, we followed the seemingly common practice in the certified training community of tuning hyperparameters on the test set (see, *e.g.*, Mao et al., 2025; Shi et al., 2021). To assess the extent to which this clearly problematic practice may inflate reported performance, we additionally performed experiments in which we tuned hyperparameters on a validation set and evaluated performance on the test set.

We considered the *CNN7* architecture of Shi et al. (2021), the *de facto* standard architecture for evaluating certified training methods (see, *e.g.*, De Palma et al., 2024b; Müller et al., 2023). We present results on CIFAR-10 (Dosovitskiy et al., 2021) for $\epsilon$-radii $\frac{2}{255}$ and $\frac{8}{255}$ and on Tiny ImageNet (Le & Yang, 2015) for $\epsilon = \frac{1}{255}$, following the general evaluation protocol of De Palma et al. (2024b) (see Appendix E). We set $d_{\text{min}} = 0.05$ to filter redundant configurations and restrict the optimisation process to configurations meeting minimum certified and natural accuracies of $40\%$ and $60\%$ for CIFAR-10 ($\epsilon = \frac{2}{255}$), $25\%$ and $40\%$ ($\epsilon = \frac{8}{255}$), and $15\%$ and $20\%$ for Tiny ImageNet. Furthermore, we chose to run CROWN-IBP without loss fusion on CIFAR-10, since this resulted in generally superior performance. Additional results on MNIST (LeCun, 1998) and on different architec-

---

[1] http://github.com/ada-research/CTRAIN

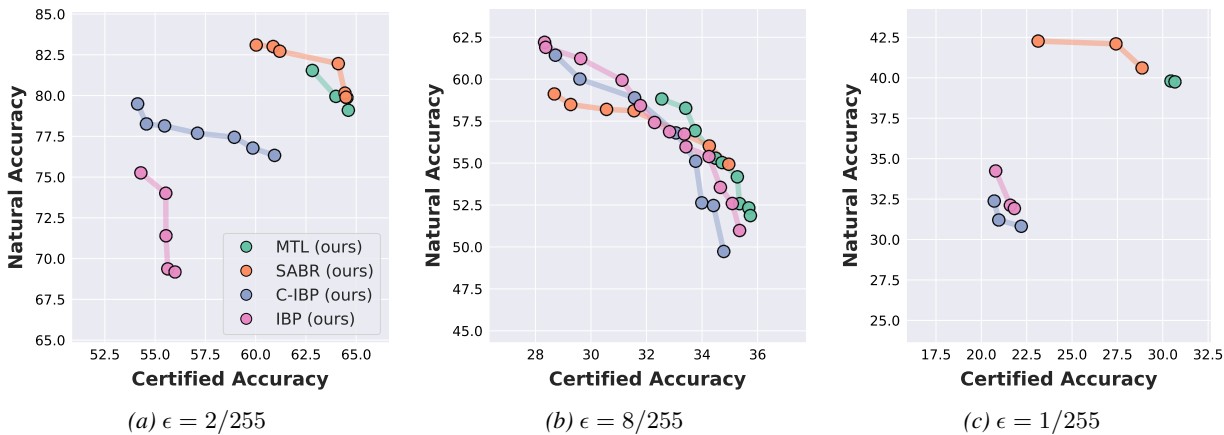

*Figure 2.* Comparison of Pareto fronts from our novel evaluation procedure on CIFAR-10 with (a) $\epsilon = \frac{2}{255}$, (b) $\epsilon = \frac{8}{255}$ and Tiny ImageNet with (c) $\epsilon = \frac{1}{255}$. The fronts enable a nuanced assessment, showing, *e.g.*, that IBP is state of the art in (b) when prioritising natural accuracy and that SABR and MTL-IBP are complementary in (c) and, to a lesser extent, in (a).

tures, including a wider *CNN7* used by Mao et al. (2024), are provided in Appendix D.

**Assessing the state of the art.** The Pareto fronts obtained from our novel multi-objective evaluation procedure allow for a more nuanced and multi-faceted assessment of the current state of the art in certified training. Instead of comparing single configurations, it is now possible to evaluate the quality of feasible solutions across the entire trade-off space. We show the Pareto fronts of all methods per benchmark in Figure 2. To assess the state of the art in IBP-based certified training, we combined all configurations found by the multi-objective hyperparameter optimisation into one single Pareto front per dataset and perturbation radius and analysed which methods contribute to this combined front. For completeness and comparability with prior work, we present a tabular comparison with previously known configurations in Appendix C.

Regarding the CIFAR-10 dataset with $\epsilon = \frac{2}{255}$, the combined Pareto set consists of networks trained with MTL-IBP and SABR. Our analysis reveals that SABR generally achieves the highest clean accuracies while maintaining strong certified robustness. However, MTL-IBP can still achieve similar certifiable guarantees once clean accuracy decreases and can achieve the strongest certifiable guarantees overall, although by a small margin. For the higher perturbation radius of $\epsilon = \frac{8}{255}$, we found that all investigated training methods contribute to the combined front. More specifically, networks trained with IBP and CROWN-IBP exhibit the strongest certifiable guarantees for higher clean accuracies, while SABR and MTL-IBP achieve better trade-offs for higher certified accuracies. This shows that IBP is a state-of-the-art method when higher natural accuracies are desired, while MTL-IBP and SABR are state of the art when prioritising certifiable guarantees. However, we

found that performance differences between methods are relatively modest across the entire trade-off space, with all approaches achieving broadly comparable results. Lastly, on Tiny ImageNet, the Pareto front includes networks trained using SABR and MTL-IBP. Here, SABR excels at increased natural accuracies, while MTL-IBP performs best when higher certified accuracies are desired.

**Comparison with the literature.** Beyond offering a new perspective on evaluating certified training techniques, we also sought to identify potentially biased or under-tuned results in prior work, by systematically comparing them against outcomes obtained with our unbiased and standardised tuning procedure. In Figure 3, we display the Pareto fronts discovered using our evaluation procedure against those from the literature, including the original publications of each method and the recent CTBench benchmark (Mao et al., 2025). In nearly all scenarios, the results from the literature are Pareto-dominated by the configurations uncovered using our novel evaluation approach.

Most notably, on CIFAR-10 with $\epsilon = \frac{2}{255}$, SABR achieves a gain of more than $1\%$ in terms of clean and certified accuracy, surpassing prior results known for this benchmark. Furthermore, our results demonstrate that MTL-IBP can achieve strong certified and clean performance at the same time, contrary to the results presented by Mao et al. (2025), where comparable certified accuracies could only be achieved for lower natural accuracies. For CROWN-IBP and IBP, we found that these older methods remain competitive, with CROWN-IBP achieving nearly a $6\%$ improvement in clean accuracy over best results from the literature.

While for $\epsilon = \frac{8}{255}$, our optimisation often did not outperform previously known results regarding certified accuracy, it found configurations with comparable certified but higher

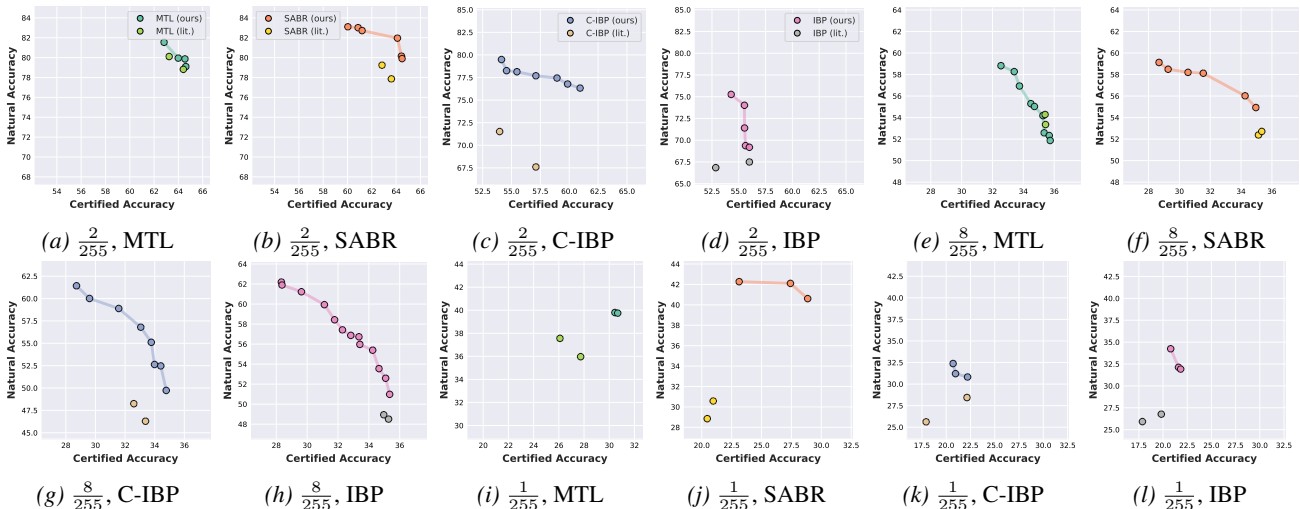

*Figure 3.* Results for CIFAR-10 for $\epsilon = \frac{2}{255}$ are shown in (a)-(d), for $\epsilon = \frac{8}{255}$ in (e)-(h), and for Tiny ImageNet for $\epsilon = \frac{1}{255}$ in (i)-(l). We compare Pareto fronts obtained using our evaluation to results given in the original publications and CTBench (Mao et al., 2025).

natural accuracy. Mao et al. (2025) suggest that all investigated methods converge to the same certified accuracy at this larger perturbation radius. We validate this result but show that the performance differences regarding clean accuracy are much less pronounced than previously assumed.

For Tiny ImageNet, we obtain new state-of-the-art results that substantially surpass prior work, with MTL-IBP achieving an improvement of about 2% in terms of clean and certified accuracy. We further demonstrate that SABR can achieve comparable results, with even stronger performance on clean data.

**Validation set tuning.** In order to assess the impact of tuning directly on the evaluation set on performance, we reran the multi-objective hyperparameter optimisation for CIFAR-10, $\epsilon \in \{\frac{2}{255}, \frac{8}{255}\}$ on a randomly selected validation set comprising 20% of the train set. We then analysed the resulting fronts obtained using incomplete verification. In Figure 4, we display a comparison for CIFAR-10, $\epsilon = \frac{2}{255}$ and provide the remaining results in Appendix D.3. Perhaps unsurprisingly, the Pareto fronts yielded through validation set tuning were always strictly dominated by those obtained when tuning on the test set directly. Interestingly, the fronts exhibited visually similar shapes, indicating that, while absolute performance degrades, relative performance characteristics may remain unchanged. These results demonstrate that previous evaluations substantially overestimated performance and do not accurately reflect the generalisation capabilities of IBP-based certified training.

**Computational costs.** Since hyperparameter optimisation for certified training as well as complete verification are inherently expensive procedures, the proposed evaluation protocol incurs substantial computational cost, which we analyse in detail in Appendix F. To reduce the computa-

tional burden, we studied the effect of reduced optimisation trials and verification timeouts on the resulting Pareto fronts. While good approximations were often obtained after 50 trials per seed, optimisation continued to improve throughout the full budget. In contrast, reducing the verification timeout for complete methods from 1000 s to 100 s on CIFAR-10 preserved the final Pareto fronts while reducing compute by more than an order of magnitude. For example, the total verification time of MTL-IBP and SABR on CIFAR-10 with $\epsilon = \frac{2}{255}$ decreased from 1311 to 208 hours and from 1585 to 227 hours, respectively. On Tiny ImageNet, slightly larger timeouts of 250 s were required to maintain identical fronts. Overall, these results suggest that verification times can be reduced substantially with little impact on the final Pareto fronts, whereas reducing the optimisation budget is considerably more detrimental to their quality.

# 6. Hyperparameter Importance Analysis

We analysed which hyperparameters most strongly influence certified and natural accuracy, in order to explain why the discovered configurations outperform previously reported ones, and to derive actionable insights for the certified training community. To this end, we used fANOVA (Hutter et al., 2014) to quantify the importance of individual hyperparameters during the optimisation procedure. Here, the importance of a hyperparameter is defined as the fraction of the variance in the predicted performance that can be attributed to it. Intuitively, changing an important hyperparameter is expected to have a large effect on performance. In Appendix G, we report the five most important hyperparameters and their importance scores for each benchmark, method and objective, *i.e.*, clean and certified accuracy, within parallel coordinate plots showing the hyperparameter values of all configura-

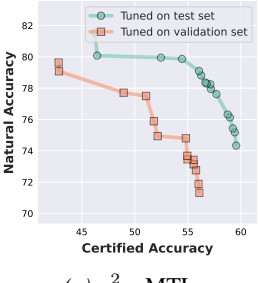
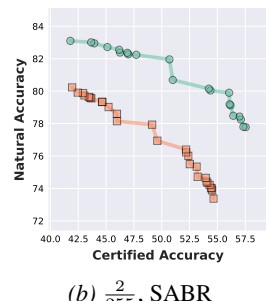
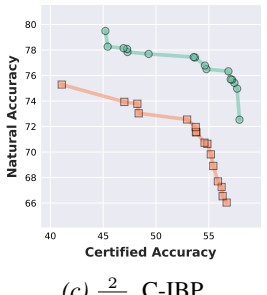
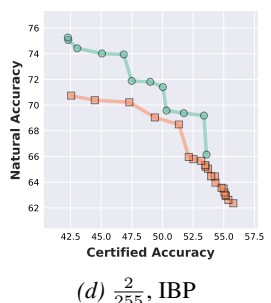

*(a)* $\frac{2}{255}$, MTL      *(b)* $\frac{2}{255}$, SABR      *(c)* $\frac{2}{255}$, C-IBP      *(d)* $\frac{2}{255}$, IBP

*Figure 4.* Comparison of Pareto fronts on CIFAR-10 with $\epsilon = \frac{2}{255}$, obtained using incomplete verification when hyperparameters are tuned on a validation set versus directly on the test set. In all cases, validation-tuned Pareto fronts are strictly dominated by those obtained via test-set tuning, indicating that prior evaluations overestimate generalisation performance.

tions in the Pareto set. In the following, we describe the most important observations made from those results.

**IBP.** Our analysis revealed that IBP yields stronger trade-offs when more time is spent on optimising for clean cross-entropy loss than done in related work. This is exemplified in $\kappa_{\text{start}}$ and $\kappa_{\text{end}}$ being highly important parameters across all scenarios, often taking larger values and interacting with longer ramp- and warm-up phases than used previously. Interestingly, for CIFAR-10 with $\epsilon = \frac{8}{255}$ and for Tiny ImageNet with $\epsilon = \frac{1}{255}$, scaling the training $\epsilon$ is also highly influential in controlling the trade-off, with larger values yielding stronger certified accuracy.

**CROWN-IBP.** Regarding CROWN-IBP, we observed a similar trend, where $\kappa$ parameters play an important role in achieving strong performance across all scenarios. This shows that the $\kappa$ transition is indeed crucial for CROWN-IBP and should not have been abolished. Interestingly, on CIFAR-10 $\beta_{\text{end}}$ is a key driver of certified accuracy, but its optimal value depends on $\epsilon$: higher values should be chosen when $\epsilon = \frac{2}{255}$, while near-zero values are essential for $\epsilon = \frac{8}{255}$. Interestingly, on the Tiny ImageNet dataset, reducing the weight of the regularisation by Shi et al. (2021) compared to prior work yields strictly better trade-offs.

**SABR.** When investigating the results for SABR, it became apparent that the subselection ratio $\tau$ is extremely effective at governing the trade-off between certified and clean accuracy, being a highly important parameter across all scenarios. Furthermore, parameters of the attack are also highly important, such as the number of optimisation steps or the scaling factor of the $\epsilon$ applied during the attack. Interestingly, using a training $\epsilon$ scaled by a factor greater than 1 is another influential mechanism for steering the robustness–accuracy trade-off across all investigated benchmarks. Crucially, SABR achieves stronger natural accuracies than previously reported for small $\epsilon$ values on CIFAR-10 and Tiny ImageNet, where longer ramp-up phases on CIFAR-10 lead to stronger results, while, interestingly, for Tiny

ImageNet, shorter ramp-up phases improve performance.

**MTL-IBP.** Lastly, we focus on the hyperparameter configurations of MTL-IBP. The method-inherent trade-off parameter $\alpha$ is very important across all scenarios, effectively steering the trade-off between natural and certified accuracy, and on many benchmarks, the scaling of the $\epsilon$ values used during training and the conducted attack also play a key role in controlling this balance. Interestingly, better trade-offs on CIFAR-10 with $\epsilon = \frac{8}{255}$ could be traced back to a higher number of warm-up epochs employed in our configurations compared to prior work. On the Tiny ImageNet benchmark, where our evaluation method uncovered the most pronounced improvements, scaled $\epsilon$ values and a smaller $\alpha$ value were the most important deviations from previous configurations.

## 7. Discussion

Our empirical evaluation reveals several important insights that challenge common assumptions in certified training and provide concrete directions for future research.

Firstly, we found that seminal methods, such as IBP and CROWN-IBP, achieve substantially stronger performance than previously reported. This implies that prior evaluations and comparisons underexplored the potential of these methods due to poor hyperparameter tuning, including the abolishment of the transition from clean to certified losses during the ramp-up phase, which proved crucial in our evaluation to achieve best trade-offs.

Secondly, our evaluation shows that performance of IBP-based certified training methods at higher perturbation radii has stalled, with older methods also contributing to the overall Pareto front of best-performing methods. Thus, future research should focus on those larger radii, where improvements remain limited. At smaller radii across both datasets, our analysis revealed that the potential of the recent methods SABR and MTL-IBP had not been fully realised, especially regarding performance on clean data.

Lastly, our analysis uncovered complementary strengths between methods, where SABR was seen to achieve stronger clean accuracy, whilst MTL-IBP provided the strongest certified guarantees at those smaller $\epsilon$-radii. Recognising these complementary strengths allows researchers and practitioners to select methods tailored to their specific objectives along the trade-off spectrum.

Beyond these observations, our study provides several actionable contributions for the certified training community. By applying a standardised multi-objective evaluation framework, we have demonstrated that prior literature often reports non Pareto-optimal configurations and that many conclusions drawn from single-configuration comparisons should be revisited.

Whilst our protocol may appear computationally costly, we used evaluation budgets comparable to prior work (Mao et al., 2025) and demonstrate in Appendix F that much smaller timeouts for complete verification suffice to determine the Pareto fronts. Nevertheless, our work reiterates that fair evaluation of IBP-based certified training methods requires considerable computational resources, potentially limiting the research community to groups with access to large-scale computing infrastructure. We therefore advocate refocusing IBP-based certified training methods toward *cheap* verification, rather than maximising the number of additionally verifiable instances through increasingly large verification timeouts (see, *e.g.*, Müller et al., 2023; De Palma et al., 2024b). This is underlined by our finding that substantially smaller cutoff times are sufficient to distinguish performance differences between methods.

Finally, although we evaluate a broad range of established datasets, architectures, and certified training methods, our findings may not directly generalise to novel methods, other datasets, or more complex threat models beyond $\ell_\infty$ perturbations. Nevertheless, we believe that our central contribution, *i.e.* advocating evaluation across the full trade-off space via standardised multi-objective hyperparameter optimisation, will remain valuable as IBP-based certified training advances toward more practically relevant benchmarks.

To summarise, we sincerely hope that our results can serve as a starting point for future comparisons to the state of the art and that our protocol will be useful for the evaluation of novel certified training methods. Furthermore, we showed that many important hyperparameters that control the trade-off have been overlooked in the past, *e.g.*, the $\kappa$ parameter in IBP and CROWN-IBP, the scaling of the $\epsilon$ value used during training and the PGD attack and the length of warm- and ramp-up phases. While many prior studies simply adopted standard values for these parameters, our analysis revealed that best trade-offs can only be achieved by exploiting complex interactions between several relevant hyperparameters. To find these configurations, state-of-the-art

Bayesian multi-objective hyperparameter optimisation techniques are required and should be consistently employed, including the adaptations presented in this work that render the optimisation process tractable.

## 8. Conclusions

In this work, we have introduced a standardised multi-objective evaluation protocol for IBP-based certified training and used it to reassess the current state of the art across widely used benchmarks and perturbation radii. By systematically exploring the hyperparameter space using multi-objective Bayesian hyperparameter optimisation and evaluating entire Pareto fronts rather than individual configurations, we obtained a more complete and reliable picture of the state of the art in certified training.

Our results demonstrate that the well-known IBP and CROWN-IBP methods perform substantially better than previously reported when properly tuned, indicating that prior comparisons have underestimated their potential. In addition, we found that progress at larger perturbation radii remains limited, with only marginal performance improvements since the inception of IBP-based certified training. However, improvements at smaller radii across CIFAR-10 and Tiny ImageNet are more pronounced than previously assumed, especially regarding performance on clean data. Importantly, our analysis has revealed that the state of the art is comprised of several complementary methods: SABR is best suited for achieving high natural accuracy, whereas MTL-IBP consistently delivers the strongest certified guarantees at smaller perturbation radii, and all investigated methods contribute to a combined Pareto front at larger radii on CIFAR-10.

Beyond benchmarking, our work provides new insights into the mechanisms that govern certified training. We show that the trade-off between clean and certified performance is influenced by a sizeable set of interacting hyperparameters – including, but not limited to, method-specific trade-off parameters. These interactions are highly complex and could only be uncovered using a state-of-the-art hyperparameter optimisation method.

Overall, our work calls for a shift in how certified training methods are evaluated and compared. We advocate for standardised multi-objective evaluation protocols that jointly tune all hyperparameters. We believe that adopting such an approach will enable more meaningful progress and facilitate the development of certified models that are both robust and practically useful.

## Impact Statement

Our paper aims to improve the performance of certified training methods, providing a full Pareto front of well-performing configurations with different accuracy-robustness trade-offs. As certified training methods are used to obtain provably safe neural networks, we do not anticipate negative ethical implications of our work. Further, our Pareto front analysis enables a nuanced assessment of the performance of certified training techniques, thereby facilitating their responsible and informed application in practice. However, while we demonstrated the effectiveness of our method across several commonly used vision datasets, this does not guarantee its effectiveness on different benchmarks, data domains or threat models.

## Acknowledgements

The authors express their sincere gratitude to Anna Münz for detailed feedback, constructive criticism, and rigorous proof-reading, and to Henning Duwe for fruitful discussions. Computing resources were provided by the German AI Service Center WestAI. Holger H. Hoos gratefully acknowledges support through an Alexander von Humboldt Professorship in Artificial Intelligence. Finally, the authors thank the reviewers for their valuable and insightful comments, which considerably improved the final version of the paper.

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

# A. Reproducibility Statement

The code to reproduce our experiments is available as part of the open-source certified training library CTRAIN (Kaulen & Hoos, 2025). We fixed the state of the library used in this publication at branch `rethinking-evaluation-paradigms`: https://github.com/ADA-research/CTRAIN/tree/rethinking-evaluation-paradigms. In our experiments, we used popular open-source datasets which can be downloaded and preprocessed via CTRAIN. We provide additional information on the setup of our experiments in Appendix E, including hardware details, software versions, neural network architectures used and detailed configuration spaces.

# B. Additional Background and Related Work

In the following, we provide additional background for our work, covering the details of interval bound propagation as well as multi-objective hyperparameter optimisation.

## B.1. Interval Bound Propagation

Given a neural network $f_\theta : \mathbb{R}^d \mapsto \mathbb{R}^c$, $c, d \in \mathbb{N}$ that maps inputs $\mathbf{x} \in \mathbb{R}^d$ to outputs $f_\theta(\mathbf{x}) \in \mathbb{R}^c$, interval bound propagation (IBP) (Gowal et al., 2019) is concerned with obtaining sound lower bounds to the vector of logit differences $\mathbf{z}(\mathbf{x}, y) \in \mathcal{R}^c$ defined as $\mathbf{z}(\mathbf{x}, y) := f_\theta(\mathbf{x})[y] \cdot \mathbf{1} - f_\theta(\mathbf{x})$. For this, consider $f_\theta$ as the composition of $L$ linear layers $h_{1,\ldots,L}$ with $h_i(\mathbf{x}^{i-1}) = \mathbf{W}_i \cdot \mathbf{x}^{i-1} + \mathbf{b}_i$ and the ReLU activation $\sigma(\mathbf{x}) := \max(0, \mathbf{x})$, i.e., $f_\theta = h_L \circ \sigma \circ h_{L-1} \circ \cdots \circ \sigma \circ h_1$. Using interval arithmetic, the axis-aligned hyper-box $\mathcal{B}_1$ that encompasses $h_1(\mathcal{B}_\infty^\epsilon)$, with $\mathcal{B}_\infty^\epsilon = \{\mathbf{x} \mid \|\mathbf{x} - \mathbf{x}_0\|_\infty \leq \epsilon\}$, is defined to have centre $\overline{\mathbf{x}}_1 = \mathbf{W} \cdot \mathbf{x}_0$ and edge length $\delta_1 = |\mathbf{W}| \cdot \epsilon$. To approximate the reachable outputs of $\sigma(\mathcal{B}_1)$, due to the non-linearity of the ReLU function, lower and upper bounds have to be propagated separately, i.e., $\mathbf{l}_2 = \sigma(\overline{\mathbf{x}}_1 - \delta_1)$ and $\mathbf{u}_2 = \sigma(\overline{\mathbf{x}}_1 + \delta_1)$. The resulting hyper-box $\mathcal{B}_2$ has centre $\overline{\mathbf{x}}_2 = \frac{\mathbf{u}_2 + \mathbf{l}_2}{2}$ and edge length $\delta_2 = \frac{\mathbf{u}_2 - \mathbf{l}_2}{2}$. By continuing this process, we can compute a hyper-box that encompasses the reachable output set of $f_\theta$, thereby allowing for the calculation of $\underline{\mathbf{z}}(\mathbf{x}, y)$.

## B.2. Hyperparameter Optimisation

**Formal definition.** In hyperparameter optimisation, let $\mathcal{A}$ be an algorithm and $\boldsymbol{\Lambda}$ its configuration space, containing the hyperparameters and their ranges considered for optimisation. When $\mathcal{A}$ is run with a hyperparameter configuration $\lambda \in \boldsymbol{\Lambda}$, we denote it as $\mathcal{A}_\lambda$. Given a data distribution $D$ with training set $D_{\text{train}}$ and test set $D_{\text{test}}$, and $l$ performance metrics $\mathbf{m} = \{m_1, \ldots, m_l\}$, each metric evaluates the performance of $\mathcal{A}_\lambda$ trained on $D_{\text{train}}$ and tested on $D_{\text{test}}$. We assume w.o.l.g. that the optimisation goal is to maximise all

metrics. We denote the metric values of configuration $\lambda$ as

$$\mathbf{m}(A_\lambda) = \big(m_1(A_\lambda), m_2(A_\lambda), \ldots, m_l(A_\lambda)\big). \quad (5)$$

The optimisation may have *constraints* $c_1(A_\lambda), \ldots, c_k(A_\lambda)$. A configuration $\lambda$ satisfies constraint $c_i$ if, and only if, $c_i(\lambda) \geq 0$, and configurations satisfying all constraints are called *feasible*.

Let $\mathbf{m}_i := \mathbf{m}(A_{\lambda_i})$ and $\mathbf{m}_j := \mathbf{m}(A_{\lambda_j})$. For two feasible configurations $\lambda_i, \lambda_j \in \boldsymbol{\Lambda}$, we say that $\lambda_i$ *Pareto dominates* $\lambda_j$ (i.e., $\mathbf{m}_i \succ \mathbf{m}_j$) if

$$\forall k \in \{1, \ldots, l\} : m_{i,k} \geq m_{j,k} \quad \text{and}$$
$$\exists k \in \{1, \ldots, l\} : m_{i,k} > m_{j,k}. \quad (6)$$

The optimisation goal is to identify the Pareto set of non-dominated feasible configurations $\boldsymbol{\Lambda}^* \subseteq \boldsymbol{\Lambda}$, such that $\lambda \in \boldsymbol{\Lambda}^*$ iff $\nexists \lambda' \in \boldsymbol{\Lambda}$ with $\mathbf{m}(A_\lambda) \prec \mathbf{m}(A_{\lambda'})$. The corresponding Pareto front is denoted $\mathbf{M}^* = \{\mathbf{m}(A_\lambda) \mid \lambda \in \boldsymbol{\Lambda}^*\}$.

Common metrics for assessing multi-objective optimisation include the *hypervolume*, defined as the Lebesgue measure of the dominated space between a reference point $r \in \mathbb{R}^l$ and an approximate Pareto front $\mathbf{M}$; we denote it as $\text{HV}(\mathbf{M}, r)$.

**Multi-objective Bayesian optimisation.** Since many real-world problems involve multiple objectives, several approaches for multi-objective optimisation have been proposed, including evolutionary algorithms (Beume et al., 2007; Deb et al., 2002) and Bayesian optimisation (Daulton et al., 2020), the latter of which we adopt in this work. Bayesian optimisation is a surrogate-based approach that iteratively samples configurations $\lambda_1, \lambda_2 \ldots, \lambda_t$ and stores them in a dataset $\zeta$. This dataset is used to train *surrogate models* $S_1 : \hat{\Lambda} \to \mathbb{R}, S_2 : \hat{\Lambda} \to \mathbb{R}, \ldots, S_l : \hat{\Lambda} \to \mathbb{R}$, each approximating an objective $m_1, \ldots, m_l$. In addition to objective estimates, surrogates provide predictive uncertainty, typically expressed as a variance $\sigma^2$. Common choices for surrogate models include Gaussian processes (Rasmussen & Williams, 2006) and random forests (Breiman, 2001). An *acquisition function* balances exploration and exploitation, and selects the configuration with the highest acquisition value for evaluation. The dataset $\zeta$ is updated with this configuration, and the process continues until a given evaluation budget is exhausted.

In the multi-objective setting, the *expected hypervolume improvement* (EHVI) acquisition function is frequently used. Let $\mathbf{m}_\lambda := \mathbf{m}(A_\lambda, D_{\text{train}}, D_{\text{test}})$ and $\mathbf{c}_\lambda := \mathbf{c}(A_\lambda, D_{\text{train}}, D_{\text{test}})$. Given a Pareto front $\mathbf{M}$ and a new configuration $\lambda \in \boldsymbol{\Lambda}$, the hypervolume improvement is defined as

$$\text{HVI}(\mathbf{M}, \lambda) = \big(\text{HV}(\mathbf{M} \cup \{\mathbf{m}_\lambda\}) - \text{HV}(\mathbf{M})\big) \cdot \mathbb{1}[\mathbf{c}_\lambda]. \quad (7)$$

*i.e.*, the additional hypervolume gained by adding $\lambda$ to the Pareto set. The EHVI is then given by $\mathrm{EHVI}(\mathbf{M}, \lambda) = \mathbb{E}[\mathrm{HVI}(\mathbf{M}, \lambda)]$

**Related Work** Multi-objective optimisation was deployed previously in multiple AutoML scenarios. For example, Dooley et al. (2023) performed joint hyperparameter optimisation and neural architecture search of CNNs to train networks which are not only accurate but also unbiased. Hennig & Lindauer (2025) used multi-objective hyperparameter optimisation to find optimal shift neural networks that balance energy efficiency and accuracy. The popular YAHPO (Pfisterer et al., 2022) benchmark offers several multi-objective hyperparameter optimisation benchmarks for tabular machine learning. The benchmarks balance between different objectives, including accuracy, memory usage and interpretability.

## C. Single-Point Comparison to Literature

For comparability with prior works and completeness, we also provide a tabular overview of the attainable performance of the uncovered Pareto-dominant configurations in comparison to literature results in Table 1. Here, we chose two configurations from our discovered Pareto sets that compare favourably to the trade-offs chosen in the respective publications that introduced the method as well as to the recent CTBench benchmark. In most cases, the results yielded by our evaluation procedure achieve comparable certified accuracies at markedly improved natural performance. In particular, on Tiny ImageNet the found configurations almost always Pareto-dominate results from related work.

## D. Additional Experiments

In the following, we give results of experiments conducted on additional datasets and architectures.

### D.1. Additional Datasets

We evaluated our approach on MNIST (LeCun, 1998) with $\epsilon = 0.3$, following the experimental setup outlined earlier. While we left our optimisation procedure unchanged, we ran verification with a cutoff time of 300s to reduce the computational burden. Nevertheless, we believe that our results regarding certified accuracy could be further strengthened when employing cutoff times of $1\,000$ seconds as done in related work (see, *e.g.*, Mao et al., 2025; De Palma et al., 2024b). We display the resulting Pareto fronts in Figure 5. In addition, we show the Pareto fronts found using our novel evaluation method in comparison to prior results in Figure 6 and Table 2.

When assessing the state of the art using the uncovered fronts, we observed that IBP, SABR and MTL-IBP con-

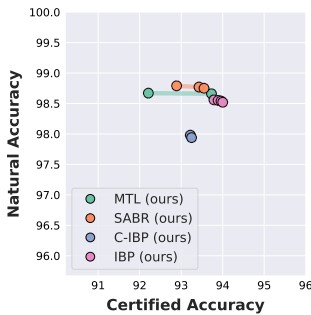

*Figure 5.* Comparison of Pareto fronts from our novel evaluation procedure on MNIST with $\epsilon = 0.3$. The fronts enable a nuanced assessment, showing that the methods MTL-IBP, SABR and IBP achieve very similar performance and, thus, contribute to the combined Pareto front over all methods.

tribute to the combined Pareto front, overall achieving very similar results. Since all methods converge to very similar trade-offs in our analysis, this suggests that a performance barrier has likely been reached on that benchmark. We therefore urge the community to interpret reported advancements on this benchmark cautiously, as properly tuned basic methods already achieve strong results. Thus, only truly significant improvements should be highlighted in the future.

While our method generally achieves comparable performance to configurations reported in the literature, it did not identify configurations that substantially surpass prior results. As outlined previously, we attribute this to the fact that current certified training techniques have likely already been tuned to the maximal performance achievable with IBP-based training for the given benchmark. However, the fact that we were able to retrieve these high-performing configurations underlines the effectiveness of our evaluation method once more.

### D.2. Additional Architectures

Recently, Mao et al. (2024) showed, both theoretically and empirically, that architecture, specifically network depth and width, has a major impact on the performance of certified training techniques. The authors found that the CNN7 Wide network exhibits optimal depth and width for certified training techniques. We investigated whether this claim still holds when considering a Pareto front as the performance measure by conducting our novel evaluation procedure on CNN5, CNN7 Wide, CNN7 Narrow and CNN9 using the CIFAR-10 dataset with $\epsilon = \frac{2}{255}$. Here, we opted for a preliminary experiment where we ran complete verification with a smaller cutoff time of 300s. We present the resulting Pareto fronts in Figure 7. Our analysis reveals that, indeed, the CNN7 Wide architecture yields very strong trade-offs across the performance space. However, there are also other architectures that contribute to a combined Pareto

*Table 1.* Comparison of the results reported from the literature to the results achieved by using our novel optimisation procedure. For each result from the literature, we selected a configuration from the Pareto front that achieves similar or better performance. Boldface marks results surpassing prior work; underlined values indicate similar performance ($\pm 0.5$). Our method typically yields configurations with higher clean accuracy and, in many cases, improved certified accuracy.

| Dataset | $\epsilon$ | Method | Source | Clean Acc. [%] (Lit.) | Cert. Acc. [%] (Lit.) | Clean Acc. [%] (ours) | Cert. Acc. [%] (ours) |
|---|---|---|---|---|---|---|---|
| CIFAR-10 | $\frac{2}{255}$ | MTL-IBP | De Palma et al. (2024b) | 80.11 | 63.24 | 79.95 | **63.99** |
| | | | Mao et al. (2025) | 78.82 | 64.41 | **79.87** | 64.54 |
| | | SABR | Müller et al. (2023) | 79.24 | 62.84 | **81.96** | **64.11** |
| | | | Mao et al. (2025) | 77.86 | 63.61 | **80.15** | **64.44** |
| | | IBP | Shi et al. (2021) | 66.84 | 52.85 | **71.40** | **55.54** |
| | | | Mao et al. (2025) | 67.49 | 55.99 | **69.18** | 55.99 |
| | | CROWN-IBP | Zhang et al. (2020) | 71.52 | 53.97 | **76.78** | **59.86** |
| | | | Mao et al. (2025) | 67.60 | 57.11 | **76.33** | **60.94** |
| | $\frac{8}{255}$ | MTL-IBP | De Palma et al. (2024b) | 53.35 | **35.44** | **55.29** | 34.49 |
| | | | Mao et al. (2025) | 54.28 | 35.41 | 54.18 | 35.27 |
| | | SABR | Müller et al. (2023) | 52.38 | 35.13 | **54.93** | 34.96 |
| | | | Mao et al. (2025) | 52.71 | **35.34** | **56.06** | 34.26 |
| | | IBP | Shi et al. (2021) | 48.94 | 34.97 | **52.59** | 35.09 |
| | | | Mao et al. (2025) | 48.51 | 35.28 | **50.98** | 35.35 |
| | | CROWN-IBP | Zhang et al. (2020) | 46.29 | 33.38 | **55.11** | 33.77 |
| | | | Mao et al. (2025) | 48.25 | 32.59 | **52.47** | **34.41** |
| Tiny ImageNet | $\frac{1}{255}$ | MTL-IBP | De Palma et al. (2024b) | 37.56 | 26.09 | **39.80** | **30.45** |
| | | | Mao et al. (2025) | 35.97 | 27.73 | **39.75** | **30.67** |
| | | SABR | Müller et al. (2023) | 28.85 | 20.46 | **40.61** | **28.86** |
| | | | Mao et al. (2025) | 30.58 | 20.96 | **42.10** | **27.42** |
| | | IBP | Shi et al. (2021) | 25.92 | 17.87 | **34.24** | **20.79** |
| | | | Mao et al. (2025) | 26.77 | 19.82 | **32.12** | **21.60** |
| | | CROWN-IBP | Xu et al. (2021) | 25.62 | 17.93 | **32.38** | **20.71** |
| | | | Mao et al. (2025) | 28.44 | 22.14 | **30.82** | 22.20 |

*Table 2.* Comparison of the results reported from the literature to the results achieved by using our novel evaluation procedure on MNIST with $\epsilon = 0.3$. For each result from the literature, we selected a configuration from the Pareto front that achieves similar or better performance. Boldface marks results surpassing prior work; underlined values indicate similar performance ($\pm 0.5$).

| Dataset | $\epsilon$ | Method | Source | Clean Acc. [%] (Lit.) | Cert. Acc. [%] (Lit.) | Clean Acc. [%] (ours) | Cert. Acc. [%] (ours) |
|---|---|---|---|---|---|---|---|
| MNIST | 0.3 | MTL-IBP | De Palma et al. (2024b) | 98.80 | 93.62 | 98.66 | 93.73 |
| | | | Mao et al. (2025) | 98.74 | 93.90 | 98.66 | 93.73 |
| | | SABR | Müller et al. (2023) | 98.75 | 92.98 | 98.77 | 93.43 |
| | | | Mao et al. (2025) | 98.66 | 93.68 | 98.75 | 93.55 |
| | | IBP | Shi et al. (2021) | 97.67 | 93.10 | **98.55** | **93.89** |
| | | | Mao et al. (2025) | 98.54 | 93.80 | 98.52 | 94.00 |
| | | CROWN-IBP | Xu et al. (2021) | 98.18 | 92.98 | 97.98 | 93.22 |
| | | | Mao et al. (2025) | **98.48** | **93.90** | 97.94 | 93.25 |

front over all architectures. For MTL-IBP and CROWN-IBP, the Pareto front also includes one CNN5 model. For SABR, more than 50% of the Pareto front consist of the standard CNN7 architecture, particularly for configurations targeting higher certified accuracies. We made similar observations for MTL-IBP, where three CNN7 models were part of the Pareto front. Finally, for standard IBP training, a single CNN7 model appears on the Pareto front, achieving a trade-off comparable to that of the CNN7 Wide models. This preliminary experiment highlights that our Pareto front analysis may reveal previously unknown performance complementarities regarding different architectures and motivates future work. However, it must be noted that the reduced cutoff time of complete verification may favour smaller models and, thus, wider models may outperform other architectures more when a higher timeout is chosen.

### D.3. Validation Set Tuning

In Figure 8 we present the resulting Pareto fronts when tuning on a validation set versus tuning on the test set directly for CIFAR-10, $\epsilon = \frac{8}{255}$. Again, we observed that the Pareto fronts obtained when tuning on the test set strictly dominate the Pareto fronts that result from tuning on the validation set and that the Pareto fronts exhibit visually similar shapes.

### D.4. Additional Certified Training Techniques

In Figure 9, we present results of CC-IBP and Exp-IBP in comparison to MTL-IBP on CIFAR-10 for $\epsilon \in \{\frac{2}{255}, \frac{8}{255}\}$. Our results show that MTL-IBP and CC-IBP perform similarly well for $\epsilon = \frac{2}{255}$, with MTL-IBP excelling for larger certified accuracies and CC-IBP performing better for higher natural accuracies. Exp-IBP performed slightly worse on that benchmark. For the larger perturbation radius, all methods performed roughly the same, but MTL-IBP yielded strongest results while dominating the two other methods.

### D.5. Comparison to Implementations from the Literature

To ensure that our reported performance gains are not due to the different codebase used for the experiments, we train with configurations reported in the literature as best-performing using CTRAIN (Kaulen & Hoos, 2025). For this, we consider configurations for SABR and MTL-IBP from their original publications and configurations for IBP and CROWN-IBP from Shi et al. (2021). We trained those on CIFAR-10 with $\epsilon = \frac{2}{255}$ and evaluated them using incomplete verification, *i.e.*, CROWN (Zhang et al., 2018). We also provide adversarial accuracy as an upper bound to the certified robustness achievable through complete verification. In Table 3 we compare the obtained results to those reported in the literature. It is important to note, that the authors of SABR did not provide results on incomplete verification (Müller et al., 2023) and Shi et al. (2021) did not provide results on CROWN-IBP. Thus, we compare to results obtained using complete verification on SABR and to results without the improvements by (Shi et al., 2021) and a longer training schedule on CROWN-IBP. The experiment shows that CTRAIN achieves similar results to the ones reported by the original authors with negligible differences. Therefore, we conclude that the success of our method cannot be attributed to the used codebase and might thus generalise to other implementations as well.

### D.6. Quality of the Proxy Objective

To make our evaluation procedure computationally efficient, we employed certified accuracy obtained through incomplete verification as a proxy for the true objective of certified accuracy assessed using costly complete verification. To assess the quality of this proxy objective, we provide Spearman rank correlation coefficients of the certified accuracies obtained using incomplete and complete verification

*Table 3.* Comparison of performances on CIFAR-10, $\epsilon = \frac{2}{255}$, of well-performing configurations reported by the respective authors across different codebases. We retrain all configurations using CTRAIN (Kaulen & Hoos, 2025) and compare them to the results reported in the original publications. CTRAIN achieves similar results across all methods, revealing that advancements achieved by our method cannot be traced back to the employed implementation.

| Dataset | $\epsilon$ | Method | Source | Clean Acc. [%] (Lit.) | Cert. Acc. [%] (Lit.) | Clean Acc. [%] (CTRAIN) | Cert. Acc. [%] (CTRAIN) | Adv. Acc. [%] (CTRAIN) |
|---|---|---|---|---|---|---|---|---|
| CIFAR-10 | $\frac{2}{255}$ | MTL-IBP | De Palma et al. (2024b) | 80.11 | 51.35 | 80.04 | 50.09 | 68.76 |
| | | SABR | Müller et al. (2023) [⋆] | 79.24 | 62.84 | 79.66 | 46.29 | 64.06 |
| | | IBP | Shi et al. (2021) | 66.84 | 52.85 | 67.35 | 53.21 | 57.50 |
| | | CROWN-IBP | Zhang et al. (2020) [†] | 71.52 | 53.97 | 67.26 | 53.97 | 57.82 |
| | $\frac{8}{255}$ | MTL-IBP | De Palma et al. (2024b) | 53.35 | 34.64 | 54.34 | 32.33 | 38.06 |
| | | SABR | Müller et al. (2023) [⋆] | 52.38 | 35.13 | 51.67 | 34.47 | 38.77 |
| | | IBP | Shi et al. (2021) | 48.94 | 34.97 | 48.04 | 33.63 | 36.93 |
| | | CROWN-IBP | Zhang et al. (2020) [†] | 46.29 | 33.38 | 46.83 | 33.13 | 35.68 |

[⋆]: Results were obtained with complete verification.
[†]: Results were obtained without improvements by Shi et al. (2021), a longer training schedule and a different architecture.

over all configurations in the Pareto front per benchmark and $\epsilon$ value in Table 4. Our analysis reveals that the proxy objective, overall, correlates highly with the final objective, achieving an average $\rho$ value of 0.7602 over all benchmarks. While we observed almost perfect correlation on CIFAR-10 with $\epsilon = \frac{8}{255}$, we sometimes obtained lower correlation coefficients on the remaining benchmarks. This indicates that at larger perturbation radii, the realisable improvements of complete verification are less pronounced. However, it must be noted that we conducted the analysis only for configurations that lie on the Pareto front since we only ran complete verification for those, and thus these results do not paint the full picture of the effectiveness of the proxy objective. While a proper analysis would be computationally infeasible, we argue that the effectiveness of our evaluation method demonstrates that our proxy objective captures the most important differences between configurations regarding certified accuracy and thereby enables the recovery of Pareto fronts with configurations outperforming the previous state of the art.

### D.7. Variance of Results

In this experiment we evaluate each configuration resulting from the hyperparameter optimisation procedure using three pseudo-random seeds to assess result variance. We present the outcomes in Figure 10, where each data point represents the mean and error bars indicate standard deviation.

It is important to note, that the algorithm performances resulting from this experiment do not create a Pareto front, since some of the configurations dominate others. The reason for this is the fact that we evaluate each configuration only once during the HPO procedure, which is a known practice when optimising neural network hyperparameters due to the high training cost associated with it (see, *e.g.*,

Zela et al., 2022). This setup allows "lucky" configurations to appear on the Pareto front, while "unlucky" ones may be excluded even if their average performance would place them on the front. Therefore, the hyperparameter optimisation might overfit to the chosen training seed. This issue is further compounded by the inherent non-determinism of GPU-based neural network training, which can lead to noticeable performance differences even with the same training seed. A strongly related topic to this issue is overtuning in hyperparameter optimisation, an active area of research (see, *e.g.*, Schneider et al., 2025; Nagler et al., 2024). One method to mitigate these phenomena is evaluating each configuration multiple times during optimisation and using its average performance, which is computationally infeasible given the costs of certified training.

However, we strongly believe that this does not undermine our results. Our final evaluation is performed on an unseen

| Dataset | Network | Method | $\epsilon$ | $\rho$ | $n$ |
|---|---|---|---|---|---|
| CIFAR-10 | CNN7 | IBP | $\frac{2}{255}$ | 0.7364 | 11 |
| | | CROWN-IBP | | 0.8545 | 10 |
| | | SABR | | 0.6727 | 11 |
| | | MTL-IBP | | 0.3333 | 10 |
| | | IBP | $\frac{8}{255}$ | 1.0000 | 13 |
| | | CROWN-IBP | | 1.0000 | 8 |
| | | SABR | | 0.8371 | 12 |
| | | MTL-IBP | | 0.9790 | 12 |
| Tiny ImageNet | | IBP | $\frac{1}{255}$ | 0.6167 | 9 |
| | | CROWN-IBP | | 0.7000 | 5 |
| | | SABR | | 0.3929 | 7 |
| | | MTL-IBP | | 1.0000 | 2 |

*Table 4.* Spearman rank correlation coefficients ($\rho$) of the proxy objective, *i.e.*, certified accuracy assessed using CROWN (Zhang et al., 2018), and the final objective, *i.e.*, certified accuracy obtained using complete verification over all configurations in the respective Pareto fronts.

test set when assessing the state of the art, and we consider differences significant only if they exceed $\pm 0.5$ compared to previously known results. This threshold corresponds to the maximum standard deviation observed when training each method with three seeds.

# E. Additional Details on the Setup of the Experiments

In the following, we give additional details on the setup of the conducted experiments.

## E.1. Hardware Details

Our experiments were conducted on two compute clusters. For running the evaluation on CIFAR-10, as well as calculating certified accuracy using complete verification for all datasets except MNIST we used a cluster in which each node is equipped with two Intel Xeon Platinum 8480+ with 210MB of L3 cache, four Nvidia H100 SXM GPUs, and 2TB of RAM running Rocky Linux 9. Each optimisation and verification experiment utilised 14 CPU cores, 220GB of RAM and one GPU. For running our method on Tiny ImageNet and MNIST, as well as verifying the Pareto fronts obtained on MNIST, we used a cluster in which each node is equipped with two Intel Xeon Platinum 8468 with 210MB of L3 cache, four Nvidia H100 NVL GPUs, 512GB of RAM running Rocky Linux 9. Here, each experiment utilised 24 CPU cores, 120GB of RAM and one GPU.

## E.2. Datasets

For our experiments, we employed several well-known datasets that have been used regularly within the certified training community. First, we used CIFAR-10 (Krizhevsky et al., 2009) containing RGB images (*i.e.*, three channels) of size $32 \times 32$ pixels associated to 10 classes (such as airplane, frog, ...). The dataset includes $50\,000$ training samples and $10\,000$ test samples. Tiny ImageNet (Le & Yang, 2015) is a subsampled version of the ImageNet dataset including $100\,000$ training samples and $10\,000$ test samples restricted to 200 classes. The resolution of each image is $64 \times 64$ with three channels. In additional experiments, we employed the MNIST dataset (LeCun, 1998) which contains grayscale images of size $28 \times 28$ with $60\,000$ training samples and $10\,000$ test samples. In line with previous work (see, *e.g.*, Mao et al., 2025; De Palma et al., 2024b; Müller et al., 2023; Shi et al., 2021; Xu et al., 2020), we normalised all datasets and used data augmentations when training on CIFAR-10 and Tiny ImageNet. More specifically, we augmented CIFAR-10 and Tiny ImageNet with random horizontal flips and random cropping to $32 \times 32$ pixels after 2-pixel padding for CIFAR-10, and to $64 \times 64$ pixels after 4-pixel padding for Tiny ImageNet. Lastly, we trained on the corresponding train

sets and report clean and certified accuracy on the test split (in line with, *e.g.*, Mao et al., 2025; De Palma et al., 2024b; Müller et al., 2023; Shi et al., 2021; Xu et al., 2020).

## E.3. Architectures

We use the CNN7 architecture from Shi et al. (2021) across all datasets and $\epsilon$ radii, which is the *de facto* standard architecture to evaluate certified training methods on (see, *e.g.*, Mao et al., 2025; De Palma et al., 2024b; Mao et al., 2024; Müller et al., 2023). This architecture employs BatchNorm layers (Ioffe & Szegedy, 2015) before every ReLU activation which improve the performance of certified training methods by reducing an imbalance between active and inactive neurons (Shi et al., 2021). To further evaluate the consistency of our tuning method and to investigate findings by Mao et al. (2024) regarding the influence of architecture on certified training, we included wide and narrow variants of CNN7 as defined by Mao et al. (2024) in additional experiments. We also show the performance on deeper and shallower versions of CNN7, named CNN9 and CNN5, respectively following Mao et al. (2024). The architectures are illustrated in Table 5.

## E.4. Experimental Setup

Generally, we mostly followed the experimental setup of De Palma et al. (2024b) for our hyperparameter optimisation. In the following, we give a detailed description how we ran the respective certified training methods during our novel evaluation method.

**Initialisation.** Before the start of the training procedure, the network is initialised using the technique proposed by Shi et al. (2021) that relies on a low-variance Gaussian distribution to prevent the *explosion* of IBP bounds during early training stages. This initialisation has been used in all recent works (see, *e.g.*, Mao et al., 2025; De Palma et al., 2024b; Müller et al., 2023) and was found to be generally beneficial to performance.

**Training schedule.** At the beginning of training, we employ a defined number of *warm-up* epochs where the standard cross-entropy loss is used. After that, the perturbation radius $\epsilon$ used for the calculation of (CROWN-)IBP bounds and during the PGD attack is gradually increased starting at 0 until it reaches its final value $\epsilon_{\text{train}}$ over a defined number of *ramp-up* epochs. To anneal to the final $\epsilon$ value, early works employed a linear schedule (Gowal et al., 2019; Zhang et al., 2020), but more recently a smoothed schedule was found to yield better results (see, *e.g.*, Mao et al., 2025; De Palma et al., 2024b; Müller et al., 2023; Xu et al., 2020). Here, $\epsilon$ is increased exponentially for the first 25% of ramp-up epochs and linearly thereafter. This leads to smaller $\epsilon$ values during the beginning of the training

*Table 5.* Model architectures of CNN7, CNN5 and CNN9 as defined by Shi et al. (2021) and Mao et al. (2024). For CNN7, we provide the number of filters for narrow and wide variants in the order of (Narrow CNN7— CNN7 — Wide CNN7)

*(a)* (Narrow, Wide) CNN7

| |
| --- |
| Convolutional: (32—64—128) filters of size $3 \times 3$, stride 1, padding 1 |
| Batch normalisation |
| ReLU activation |
| Convolutional: (32—64—128) filters of size $3 \times 3$, stride 1, padding 1 |
| Batch normalisation |
| ReLU activation |
| Convolutional: (64—128—256) filters of size $3 \times 3$, stride 2, padding 1 |
| Batch normalisation |
| ReLU activation |
| 2 ×    Convolutional: (64—128—256) filters of size $3 \times 3$, stride 1, padding 1 |
| Batch normalisation |
| ReLU activation |
| Linear: 512 neurons |
| Batch normalisation |
| ReLU activation |
| Linear: no. of classes in dataset |

*(b)* CNN5

| |
| --- |
| Convolutional: 64 filters of size $3 \times 3$, stride 1, padding 1 |
| Batch normalisation |
| ReLU activation |
| Convolutional: 64 filters of size $3 \times 3$, stride 2, padding 1 |
| Batch normalisation |
| ReLU activation |
| Convolutional: 128 filters of size $3 \times 3$, stride 2, padding 1 |
| Batch normalisation |
| ReLU activation |
| Linear: 512 neurons |
| Batch normalisation |
| ReLU activation |
| Linear: no. of classes in dataset |

*(c)* CNN9

| |
| --- |
| 2 ×    Convolutional: 64 filters of size $3 \times 3$, stride 1, padding 1 |
| Batch normalisation |
| ReLU activation |
| Convolutional: 128 filters of size $3 \times 3$, stride 2, padding 1 |
| Batch normalisation |
| ReLU activation |
| 4 ×    Convolutional: 128 filters of size $3 \times 3$, stride 1, padding 1 |
| Batch normalisation |
| ReLU activation |
| Linear: 512 neurons |
| Batch normalisation |
| ReLU activation |
| Linear: no. of classes in dataset |

process, which contributes to training stability. Note that the $\epsilon$ radius used during training does not need to match the $\epsilon$ value used for evaluation. In some cases, training with a larger $\epsilon$ radius than that used for evaluation has been shown to be beneficial (see, *e.g.*, Shi et al., 2021; Gowal et al., 2019). For IBP and CROWN-IBP training, we chose to include additional parameters that are annealed during the ramp-up phase. Both methods employ a $\kappa$ parameter (Zhang et al., 2020; Gowal et al., 2019) which weighs certified with clean loss, *i.e.*, $\kappa \cdot \mathcal{L}(f_\theta(\mathbf{x}), y) + (1 - \kappa) \cdot \mathcal{L}_{\text{ver}}(f_\theta(\mathbf{x}), y))$. During ramp-up $\kappa$ smoothly transitions from $\kappa_{\text{start}}$ to $\kappa_{\text{end}}$, where $\kappa_{\text{start}} \geq \kappa_{\text{end}}$. Analogously, for CROWN-IBP we included the $\beta$ parameter (Zhang et al., 2020) that additionally weighs verified losses obtained through CROWN-IBP and IBP to calculate the final verified loss used, *i.e.*, $\mathcal{L}_{\text{ver}}(\mathbf{x}, y) = \beta \cdot \mathcal{L}_{\text{CROWN-IBP}}(\mathbf{x}, y) + (1 - \beta) \cdot \mathcal{L}_{\text{IBP}}(\mathbf{x}, y)$. This parameter transitions from $\beta_{\text{start}}$ to $\beta_{\text{end}}$ with $\beta_{\text{start}} \geq \beta_{\text{end}}$. This way, tighter CROWN-IBP bounds are only employed to stabilise the beginning of the training process which may result in superior performance to using CROWN-IBP bounds throughout (Zhang et al., 2020). However, it is important to notice that by setting $\kappa_{\text{start}} = \kappa_{\text{end}} = 0$ and $\beta_{\text{start}} = \beta_{\text{end}} = 1$, experimental setups used by Shi et al. (2021) and Mao et al. (2025) can be achieved, which only employ IBP or CROWN-IBP losses respectively. After the ramp-up phase, training is carried out over the full epsilon radius until it finishes. Regarding the number of epochs, we follow De Palma et al. (2024b) and train for 70 epochs on MNIST, 160 epochs for $\epsilon = \frac{2}{255}$ and 260 epochs for $\epsilon = \frac{8}{255}$ on CIFAR-10 and 160 epochs on Tiny ImageNet.

**Regularisation.** During the ramp-up phase, we employed the regulariser proposed by Shi et al. (2021) which is composed of two terms. One that penalises the explosion of IBP bounds during training time and one that balances inactive and active ReLU activations, *i.e.*, neurons that behave only linearly and non-linearly for all inputs within the $\epsilon$ ball. The magnitude of this regularisation is controlled by two factors; a parameter $\lambda$ and a decay factor $1 - \frac{\epsilon}{\epsilon_{\text{train}}}$ with both of which the loss term is multiplied. This ensures that the regularisation is most prominently employed during the beginning of the training process which contributes to more stable training. In addition, we used $\ell_1$ regularisation weighed by a specified parameter. For its calculation, we exclusively considered the magnitude of weights in convolutional and linear layers in line with previous work (see, *e.g.*, Mao et al., 2025; De Palma et al., 2024b; Shi et al., 2021).

**Optimisation.** We included the choice of an optimiser as well as the learning rate as part of our tuning scheme. Generally, we support *Adam* (Kingma & Ba, 2015), *AdamW* (Loshchilov & Hutter, 2019) as well as *RAdam* (Liu et al., 2020). We did not tune the internal hyperparameters of the optimisers, such as their $\beta$ values and weight decay, but

used the defaults provided in PyTorch (Paszke et al., 2019). It is worth noting that prior works did not consider different optimisers but exclusively relied on Adam for the optimisation; a choice not in line with advancements in the broader ML community (see, *e.g.*, Liu et al., 2022; Wightman et al., 2021). For all conducted experiments, we employed a batch size of 512 whereas related work usually employed batch sizes of 256 on MNIST and 128 on CIFAR-10 and Tiny ImageNet (see, *e.g.*, Mao et al., 2025; De Palma et al., 2024b; Müller et al., 2023; Shi et al., 2021). While we experienced in preliminary experiments that higher batch sizes do hurt the performance of certified training, we aimed to conduct our tuning using a higher batch size to fully exploit the capabilities of modern GPUs. In addition, our method searches for two epochs after ramp-up at which the learning rate is decayed by a given factor that is also chosen as part of the hyperparameter optimisation.

**Batch normalisation layers.** Shi et al. (2021) showed that BatchNorm (Ioffe & Szegedy, 2015) layers are generally beneficial to the performance of certified training of deep neural networks. Therefore, we also employ them after every activation in the networks considered for our evaluation. In the literature, there are several options on how the statistics of the layers used to normalise batches should be set. Shi et al. (2021) and Müller et al. (2023) set the statistics based exclusively on unperturbed data, while De Palma et al. (2024b) use statistics over adversarial examples for the IBP bounds. At evaluation time, De Palma et al. (2024b) consider the statistics over both, perturbed and clean data. Mao et al. (2025) proposed to use statistics of unperturbed data for the PGD attack as well as for training. At test time, the authors employed statistics obtained over the whole population. Since multiple approaches exist and it is, to date, unclear whether any of them actually result in decisive performance differences, we chose to adopt the standard setting of CTRAIN that follows the approach of De Palma et al. (2024b) for SABR and MTL-IBP and the approach of Shi et al. (2021) for CROWN-IBP and IBP.

**Hyperparameter optimisation.** In our hyperparameter optimisation setup, we use the BoTorch (Balandat et al., 2020) sampler of Optuna (Akiba et al., 2019) with 10 initial random samples. We use a Gaussian Process as a surrogate model, with lengthscales as recommended by Hvarfner et al. (2024) and a RBF kernel. The Gaussian Process hyperparameters are optimised using L-BFGS-B with marginal log likelihood loss. The inputs to the Gaussian process are normalised to the range $[0, 1]$ and the target values are standardised. We optimise the acquisition function using L-BFGS-B. All design choices are based on the values reported by Akiba et al. (2019). Our hyperparameter optimisation method does not use any previously known configurations or priors, making the optimisation procedure generalisable for new,

unseen scenarios.

### E.5. Additional Implementation Details

To run our optimisation method, we relied on CTRAIN (Kaulen & Hoos, 2025) in version `0.4.2` for the implementation of the certified training methods. CTRAIN includes implementations of several state-of-the-art methods, including the methods investigated in this work, as well as the proposed initialisation and regularisation procedures of Shi et al. (2021). Further, it implements IBP, CROWN-IBP and CROWN (Zhang et al., 2018) for incomplete verification and the adversarial attack PGD (Madry et al., 2018) for quickly disproving robustness. For the bounding process and incomplete verification, CTRAIN in turn relies on the `auto_LiRPA` library (Xu et al., 2020) at commit `cf0169c`. Lastly, the neural network training is carried out using PyTorch (Paszke et al., 2019) in version `2.3.1`.

### E.6. Complete Verification

For complete verification, we used the state-of-the-art (Kaulen et al., 2025b; König et al., 2024) complete verification system $\alpha\beta$-CROWN (Wang et al., 2021; Xu et al., 2021; Zhang et al., 2018). While it is known that careful parameter tuning of $\alpha\beta$-CROWN is crucial to obtain strong results, we used the system in its standard configuration to not create a biased evaluation, where one certified training method or network architecture might benefit more from the selected parameter choices. We set the batch size of branch and bound domains to the highest number our hardware could accommodate, resulting in a batch size of 1024 for CNN7, Narrow CNN7 and CNN5 and a batch size of 512 for CNN7 Wide and CNN9 on CIFAR-10. We used a batch size of 1024 for MNIST and 16 for verifying networks trained on Tiny ImageNet. We used a cutoff time of $1\,000$s in wall-clock time for verification of CNN7 on CIFAR-10 and Tiny ImageNet. For MNIST and the results on additional architectures presented later, we used a cutoff of 300s in wall-clock time to keep computational demands manageable.

### E.7. Configuration Spaces

The configuration spaces used in our experiments are shown in Table 6. Each space consists of a set of base hyperparameters shared across all methods, extended with method-specific ones where necessary. In the following, we provide a brief explanation of each hyperparameter included. Generally, we ensured in our design of the search space that it encompasses all previously chosen parameter values from the literature but also includes all sensible parameter choices to allow for the discovery of novel, well-performing configurations.

**Warm up epochs** refer to the number of epochs for which the network is trained on clean cross-entropy loss at the beginning of the training schedule.

**Ramp-up epochs** refer to the previously explained training phase, where $\epsilon$ is annealed from 0 to its final value. We employ 10 such epochs at least and make the maximum number dependent on the number of total epochs the network should be trained for, thereby making the search space flexible and applicable to new benchmarks. At most, we extend the ramp-up phase through 75% of the total number of epochs. This way, the ramp-up phase will have completed at the end of training, even when the maximally allowed warm- and ramp-up durations are chosen.

**LR decay factor** describes the factor by which the learning rate is decayed at up to two epochs after the ramp-up phase, for which we also optimise.

**LR decay epoch {1,2}** describe the points in time at which the learning rate is decayed. We calculate the first point by adding *LR decay epoch 1* to the number of warm- and ramp-up epochs, ensuring that the learning rate is only decayed after the ramp-up phase completed. The second point is calculated analogously, by adding the value of *LR decay epoch 2* to the epoch at which the learning rate was decayed first. If any of these decay epochs exceed the total number of training epochs, they are ignored.

**L1 regularisation weight** refers to the weight with which L1 regularisation is employed during training.

**Shi regularisation weight** refers to the $\lambda$ parameter which refers to the magnitude of the regularisation proposed by Shi et al. (2021) during the ramp-up phase.

**Train $\epsilon$ factor** scales the $\epsilon$ value the network is evaluated on by a given factor for training. In some cases, this has shown to be beneficial (see, *e.g.*, Gowal et al., 2019; Shi et al., 2021.

**Optimiser** refers to the choice of the optimiser used for the training procedure. We include *Adam* (Kingma & Ba, 2015), *AdamW* (Loshchilov & Hutter, 2019) and *RAdam* (Liu et al., 2020).

**Learning rate** refers to the initial learning rate employed by the previously chosen optimiser.

**Start & end $\kappa$** refer to the $\kappa$ value employed in IBP and CROWN-IBP to weigh standard cross-entropy loss and the certified loss. During the ramp-up phase, $\kappa_{\text{start}}$ is gradually decreased to $\kappa_{\text{end}}$, placing greater weight on the natural loss in the early stages to stabilise training before progressively shifting the focus toward the certifiability objective. To ensure that $\kappa_{\text{start}}$ always exceeds $\kappa_{\text{end}}$, we define the latter as a multiplicative factor $c$ of the former, *i.e.*, $\kappa_{\text{end}} = \kappa_{\text{start}} \times c$ and optimise the factor $c$ instead of optimising $\kappa_{\text{end}}$ directly.

**Start & end** $\beta$ are handled analogously, but we fix $\beta_{\text{start}} = 1.0$ to ensure that the full benefit of the tighter relaxation used in CROWN-IBP is employed to stabilise early training stages.

$\tau$ refers to the subselection ratio used in SABR that weighs certified with adversarial loss (De Palma et al., 2024b; Müller et al., 2023).

$\alpha$ refers to the parameter of MTL-IBP that weighs certified with adversarial loss (De Palma et al., 2024b).

**PGD steps, step size, restarts and $\epsilon$ scaling factor** refer to the parameters of the adversarial attack employed during training to approximate the adversarial loss (Madry et al., 2018). Here, steps specify the number of optimisation steps, while step size indicates the magnitude of the input change allowed per iteration. To keep training costs tractable, we chose to always randomly initialise the attack once within the $\epsilon$ ball and not multiple times as done by Mao et al. (2025); a choice consistent with multiple other works in the field (De Palma et al., 2024b; Müller et al., 2023; Madry et al., 2018). This strategy leverages the fact that each training sample is reinitialised differently across epochs, yielding a good approximation of the worst-case adversarial loss overall. Finally, we optimise a factor that scales the $\epsilon$ radius in the adversarial attack, increasing the emphasis on the adversarial loss when combined with the certified loss (De Palma et al., 2024b), achieving a similar effect to ReLU shrinking as used by Müller et al. (2023).

## F. Computational Costs

We present the computational costs for both the hyperparameter optimisation runs and the verification in Table 7. We display the required total cost for discovery and complete verification of the Pareto front as well as the average verification time per instance. We show that hyperparameter optimisation costs scale directly with training costs, with Tiny ImageNet being the most expensive benchmark due to its larger scale. Across the same dataset, architectures with fewer parameters incur lower optimisation costs, with CNN5 being the cheapest to optimise. However, regarding the costs of complete verification, perhaps surprisingly, the highest costs occur on the CIFAR-10 dataset with $\epsilon = \frac{2}{255}$. We attribute this to the fact that, on this benchmark, complete verification methods achieve the largest improvements compared to cheaper, incomplete methods. On the other benchmarks, incomplete methods are often sufficient to certify most provably robust instances.

### F.1. Improving Efficiency

While we are aware that our conducted evaluation requires substantial computational resources, we are confident that our proposed evaluation procedure is immensely valuable even when less computing budget is available. For this, we analysed how the Pareto fronts obtained from our evaluation protocol change when using smaller cutoff times during complete verification and when running the optimisation for fewer trials per seed. We display results for varying complete verification cutoff times in Figure 11 and for varying evaluation budgets in Figure 12. We observed that on CIFAR-10, small complete verification timeouts of 100s are sufficient to recover most configurations that lie on the Pareto fronts yielded when using the timeout of 1000s. Using these smaller timeouts can reduce the required compute by more than 10x, *e.g.*, the overall verification time of MTL-IBP and SABR on CIFAR-10 with $\epsilon = \frac{2}{255}$ reduced from 1311.36 hours to 207.70 hours and from 1585.20 hours to 226.81 hours respectively. For Tiny ImageNet, slightly larger timeouts of 250s are required for an accurate approximation of the Pareto front. This resulted in reduced overall verification times of 118.84 and 444.90 hours for MTL-IBP and SABR respectively in comparison to the 207.88 and 887.92 hours required when using a timeout of 1000s. With increasing timeouts, the fronts shift to the left, *i.e.*, achieve higher certified accuracy, but do not considerably change most of the time. Thus, we conclude that when providing final results in academic papers, high complete verification timeouts are needed to achieve strongest certified accuracies, but for the discovery of configurations on the Pareto front, far smaller timeouts are sufficient. Thus, we believe that our approach provides an efficient procedure to identify best-performing configurations for practitioners and researchers.

In addition, we investigated how reduced evaluation budgets affect the discovered fronts (see Figure 12). Here, we found that, while the optimisation continues to improve the Pareto fronts until the whole evaluation budget is exhausted, good approximations of the Pareto set can be obtained after 50 trials per seed most of the time. In Table 8 we show the improvement of the hypervolume of the points on the Pareto front with respect to the reference point $(0, 0)$. This analysis supports our claim and reveals that biggest improvements are usually achieved during early stages of the optimisation. However, there exist some cases where important improvements are only uncovered after 75 trials per seed. Thus, when compute must be saved, we would advise researchers and practitioners to first reduce the timeout of complete verification, before reducing the evaluation budget of the hyperparameter optimisation since the latter has a more pronounced effect on solution quality.

## G. Hyperparameter Importance Plots

In Figures 13, 14, 15 and 16 we display parallel coordinate plots that show the 5 most influential hyperparameters per method and benchmark for the analysis conducted in Section 6.

*Table 6.* Configuration spaces employed in our hyperparameter optimisation method for certified training. Square brackets indicate continuous parameters for which we give inclusive upper and lower limits. Curly brackets indicate sets out of which the optimiser can choose one option. Finally, single numbers indicate constants, *i.e.*, parameters that remain unchanged throughout the hyperparameter optimisation.

| Method | Hyperparameter | Range |
|---|---|---|
| All | Warm up epochs | $[0, 5]$ |
| | Ramp up epochs | $[10, 0.75 \cdot \text{Total Epochs}]$ |
| | LR decay factor | $[1e\text{-}5, 0.9]$ |
| | LR decay epoch 1 | $[0, 0.5 \cdot \text{Total Epochs}]$ |
| | LR decay epoch 2 | $[0, 0.25 \cdot \text{Total Epochs}]$ |
| | L1 regularisation weight | $[1e\text{-}8, 1e\text{-}4]$ |
| | Shi regularisation weight | $[0.0, 1.0]$ |
| | Train $\epsilon$ factor | $[1.0, 2.0]$ |
| | Optimiser | $\{\text{Adam, AdamW, RAdam}\}$ |
| | Learning rate | $[1e\text{-}5, 1e\text{-}1]$ |
| IBP | Start $\kappa$ | $[0, 1]$ |
| | End $\kappa$ | $[0, 1]$ |
| CROWN-IBP | Start $\kappa$ | $[0, 1]$ |
| | End $\kappa$ | $[0, 1]$ |
| | Start $\beta$ | $1.0$ |
| | End $\beta$ | $[0, 1]$ |
| SABR | $\tau$ | $[0.001, 0.5]$ |
| | PGD steps | $[1, 10]$ |
| | PGD step size | $[0.1, 2]$ |
| | PGD restarts | $1$ |
| | PGD $\epsilon$ scaling factor | $[1, 3]$ |
| MTL-IBP | $\alpha$ | $[0.001, 0.5]$ |
| | PGD steps | $[1, 10]$ |
| | PGD step size | $[0.1, 2]$ |
| | PGD restarts | $1$ |
| | PGD $\epsilon$ scaling factor | $[1, 3]$ |

*Table 7.* Computation time of our experiments in wall-clock time. For each experiment, we show the time required for the hyperparameter optimisation, the average verification time required per instance as well as the total time used for complete verification of the Pareto front. If not indicated otherwise, we report verification times over the complete test-set with a per-instance timeout of 1 000s in wall-clock time.

| Dataset | Network | Method | $\epsilon$ | HPO (h) | Verification (s) Average | Verification (h) Total |
|---|---|---|---|---|---|---|
| CIFAR-10 $^\star$ | CNN5 | MTL-IBP | $\frac{2}{255}$ | 113.69 | 9.32 | 258.94 |
| | | SABR | | 111.02 | 10.64 | 325.01 |
| | | CROWN-IBP | | 135.15 | 6.47 | 161.72 |
| | | IBP | | 62.95 | 2.16 | 60.05 |
| CIFAR-10 | CNN7 | MTL-IBP | $\frac{2}{255}$ | 236.68 | 47.35 | 1314.97 |
| | | SABR | | 226.83 | 52.03 | 1589.94 |
| | | CROWN-IBP | | 318.13 | 27.11 | 752.00 |
| | | IBP | | 95.70 | 7.39 | 225.95 |
| CIFAR-10 | CNN7 | MTL-IBP | $\frac{8}{255}$ | 340.25 | 11.73 | 390.91 |
| | | SABR | | 296.88 | 15.54 | 517.89 |
| | | CROWN-IBP | | 457.43 | 9.10 | 202.11 |
| | | IBP | | 158.81 | 13.69 | 494.29 |
| CIFAR-10 $^\star$ | CNN9 | MTL-IBP | $\frac{2}{255}$ | 339.96 | 13.60 | 411.15 |
| | | SABR | | 336.08 | 18.91 | 682.27 |
| | | CROWN-IBP | | 434.99 | 8.15 | 158.38 |
| | | IBP | | 140.74 | 3.99 | 166.24 |
| CIFAR-10 $^\star$ | Narrow CNN7 | MTL-IBP | $\frac{2}{255}$ | 182.16 | 16.61 | 461.38 |
| | | SABR | | 172.72 | 13.95 | 465.08 |
| | | CROWN-IBP | | 211.36 | 5.29 | 190.07 |
| | | IBP | | 75.45 | 2.48 | 75.80 |
| CIFAR-10 $^\star$ | Wide CNN7 | MTL-IBP | $\frac{2}{255}$ | 409.45 | 13.98 | 266.47 |
| | | SABR | | 399.83 | 21.36 | 770.60 |
| | | CROWN-IBP | | 624.31 | 8.76 | 216.31 |
| | | IBP | | 164.31 | 4.66 | 181.33 |
| MNIST $^\star$ | CNN7 | MTL-IBP | 0.3 | 117.15 | 55.02 | 305.69 |
| | | SABR | | 104.22 | 4.79 | 93.22 |
| | | CROWN-IBP | | 140.0 | 27.05 | 300.56 |
| | | IBP | | 51.22 | 3.16 | 43.91 |
| TinyImageNet | CNN7 | MTL-IBP | $\frac{1}{255}$ | 1576.51 | 37.43 | 207.96 |
| | | SABR | | 1494.89 | 45.68 | 888.30 |
| | | CROWN-IBP | | 1567.99 | 15.12 | 209.96 |
| | | IBP | | 757.65 | 16.32 | 408.07 |

$^\star$: Selected networks were verified with a per-instance timeout of 300 seconds in wall-clock time.

*Table 8.* Hypervolume Improvement with respect to the number of trials used per seed during the multi-objective hyperparameter optimisation. In parentheses, we display the relative improvement to the previous point.

| Dataset | Method | $\epsilon$ | 10 | 20 | 50 | 75 | 100 |
|---|---|---|---|---|---|---|---|
| CIFAR-10 | IBP | 2/255 | 0.3600 | 0.3729 (+3.59%) | 0.3970 (+6.45%) | 0.3972 (+0.05%) | 0.3994 (+0.56%) |
| | CROWN-IBP | | 0.3978 | 0.4323 (+8.68%) | 0.4397 (+1.69%) | 0.4445 (+1.09%) | 0.4573 (+2.88%) |
| | SABR | | 0.4280 | 0.4596 (+7.39%) | 0.4706 (+2.38%) | 0.4744 (+0.83%) | 0.4752 (+0.17%) |
| | MTL-IBP | | 0.4478 | 0.4679 (+4.48%) | 0.4754 (+1.61%) | 0.4823 (+1.45%) | 0.4823 (+0.00%) |
| | IBP | 8/255 | 0.1829 | 0.1888 (+3.22%) | 0.2106 (+11.59%) | 0.2120 (+0.64%) | 0.2132 (+0.57%) |
| | CROWN-IBP | | 0.1427 | 0.1811 (+26.92%) | 0.1961 (+8.28%) | 0.2063 (+5.21%) | 0.2074 (+0.50%) |
| | SABR | | 0.1370 | 0.1753 (+27.95%) | 0.1886 (+7.60%) | 0.1909 (+1.18%) | 0.1936 (+1.42%) |
| | MTL-IBP | | 0.1522 | 0.1761 (+15.67%) | 0.1990 (+13.02%) | 0.2020 (+1.51%) | 0.2026 (+0.30%) |
| Tiny ImageNet | IBP | 1/255 | 0.0528 | 0.0602 (+13.92%) | 0.0698 (+15.90%) | 0.0699 (+0.18%) | 0.0702 (+0.40%) |
| | CROWN-IBP | | 0.0532 | 0.0568 (+6.63%) | 0.0637 (+12.19%) | 0.0649 (+1.87%) | 0.0659 (+1.57%) |
| | SABR | | 0.0851 | 0.0914 (+7.41%) | 0.1030 (+12.76%) | 0.1102 (+7.00%) | 0.1124 (+1.96%) |
| | MTL-IBP | | 0.1032 | 0.1074 (+4.11%) | 0.1125 (+4.68%) | 0.1129 (+0.43%) | 0.1136 (+0.57%) |

# H. Concrete Hyperparameter Configurations

We provide samples of the hyperparameter configurations found by our optimisation procedure in Tables 9, 10 and 11. As the number of configurations in the Pareto fronts can be rather high, we provide a selection of three configurations: one from each edge of the fronts (highest natural accuracy, highest certified accuracy) and one configuration from the middle of the front, selected based on the median certified accuracy.

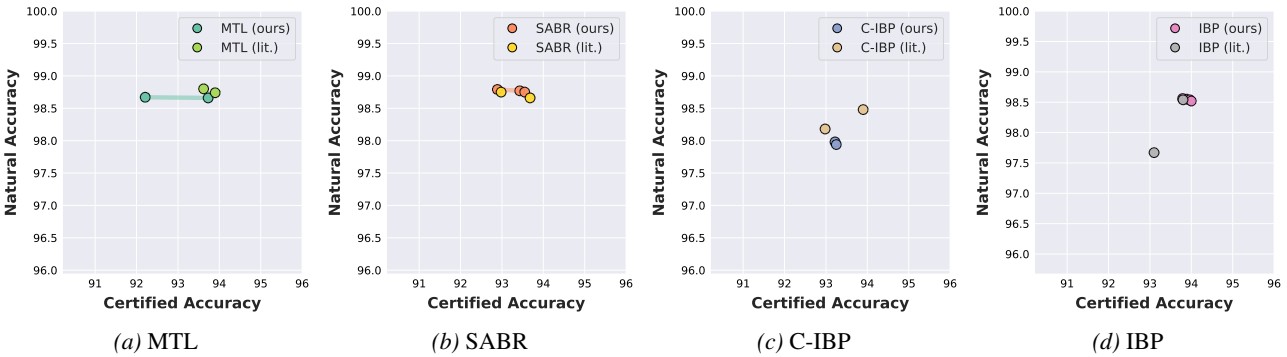

*Figure 6.* Results for MNIST with $\epsilon = 0.3$ yielded by our method. We compare Pareto fronts obtained using our method to results given in the original publications and the recent CTBench benchmark (Mao et al., 2025).

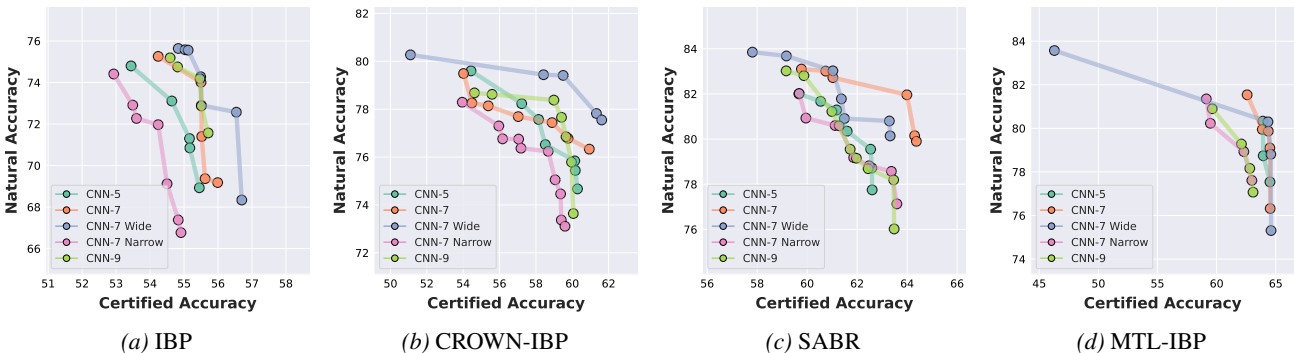

*Figure 7.* Pareto fronts on CIFAR-10 with $\epsilon = \frac{2}{255}$ yielded by our method for the architectures CNN5, CNN7, CNN7 Wide, CNN7 Narrow as well as CNN9.

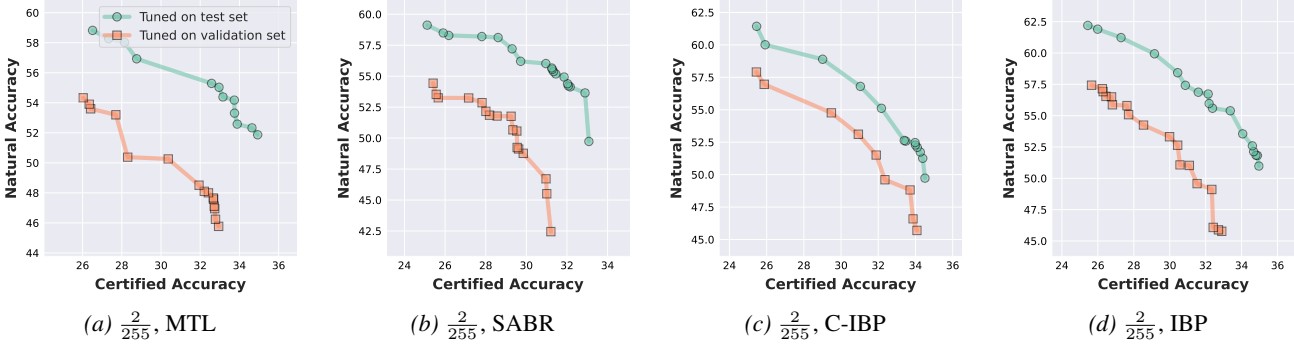

*Figure 8.* Comparison of Pareto fronts on CIFAR-10 with $\epsilon = \frac{8}{255}$, obtained using incomplete verification when hyperparameters are tuned on a validation set versus directly on the test set. In all cases, validation-tuned Pareto fronts are strictly dominated by those obtained via test-set tuning, indicating that prior evaluations overestimate generalisation performance.

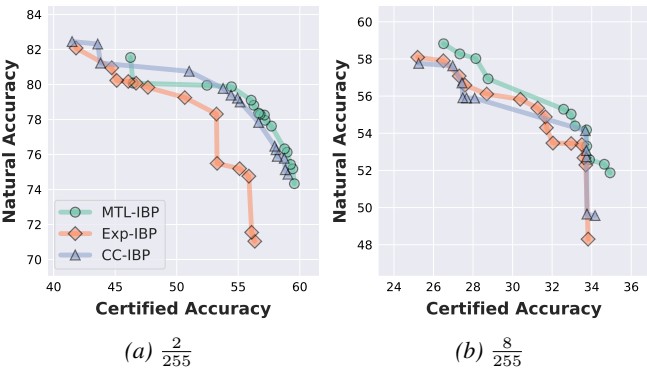

*Figure 9.* Performance of MTL-IBP, Exp-IBP and CC-IBP (De Palma et al., 2024b) on CIFAR-10, obtained using incomplete verification.

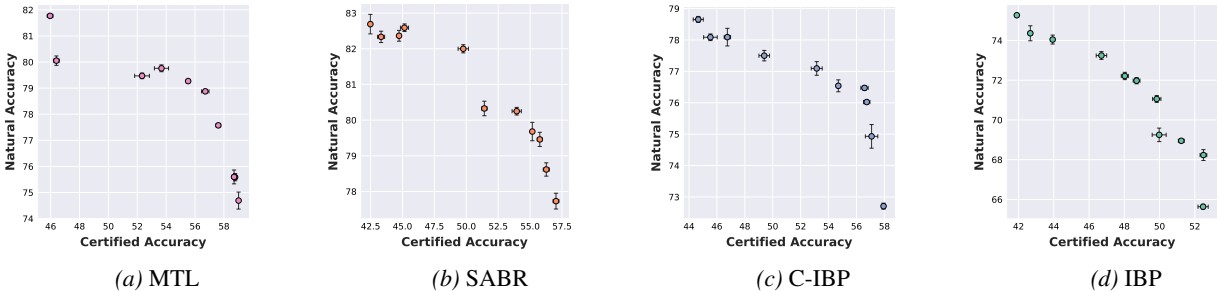

*Figure 10.* Pareto fronts obtained using our method on CIFAR-10 with $\epsilon = \frac{2}{255}$ with error bars. Each dot represents the average performance over three pseudo-random seeds and error bars indicate standard deviation.

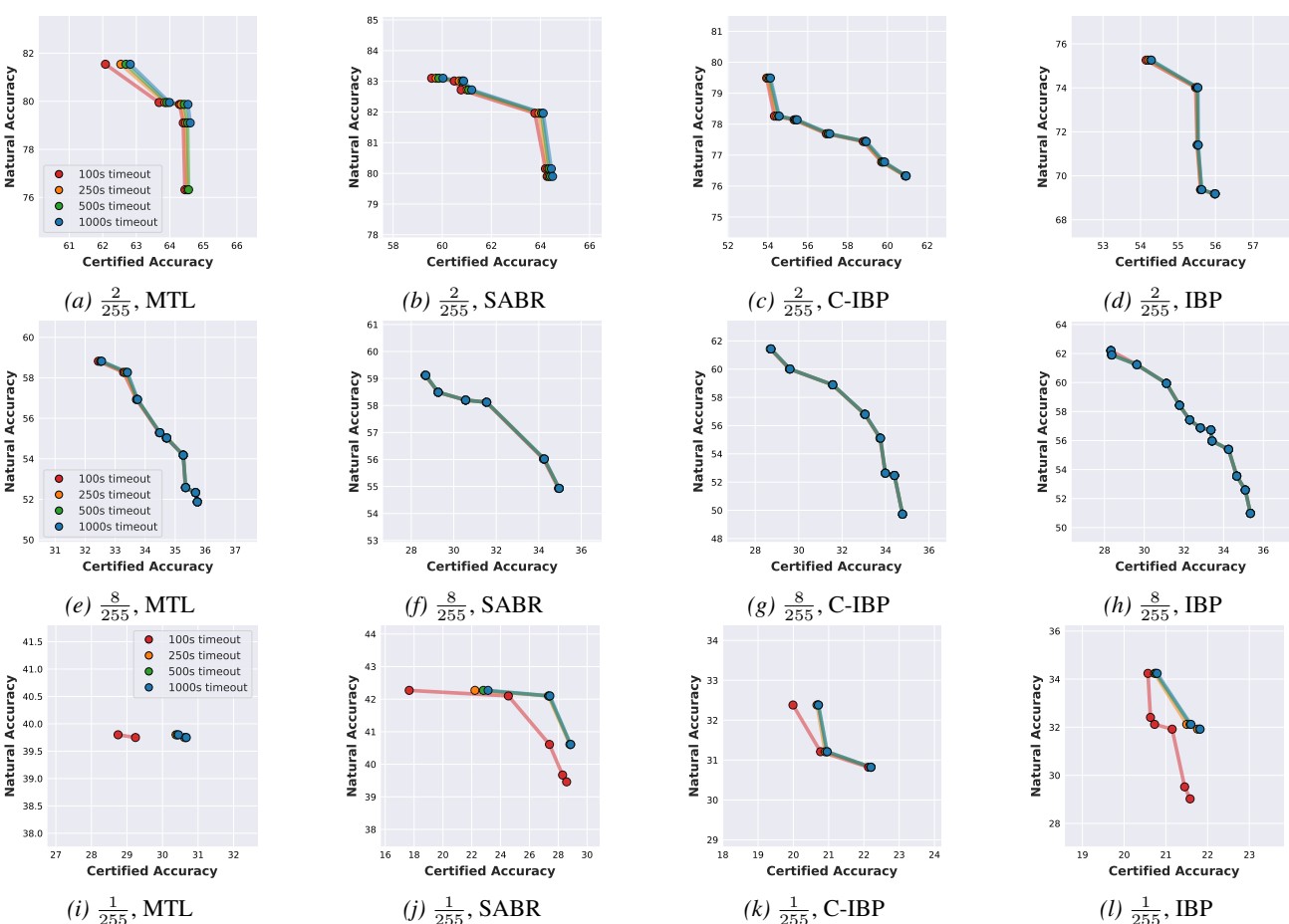

*Figure 11.* Evolution of the Pareto fronts of MTL-IBP, SABR, CROWN-IBP and IBP with varying cutoff times for complete verification. It becomes apparent that on CIFAR-10 cutoff times of 100s are sufficient to identify Pareto fronts and on Tiny ImageNet only 250s are required. Thus, we conclude that high complete verification timeouts are only needed for benchmarking purposes where best certified accuracies are crucial, but not for efficient multi-objective hyperparameter optimisation.

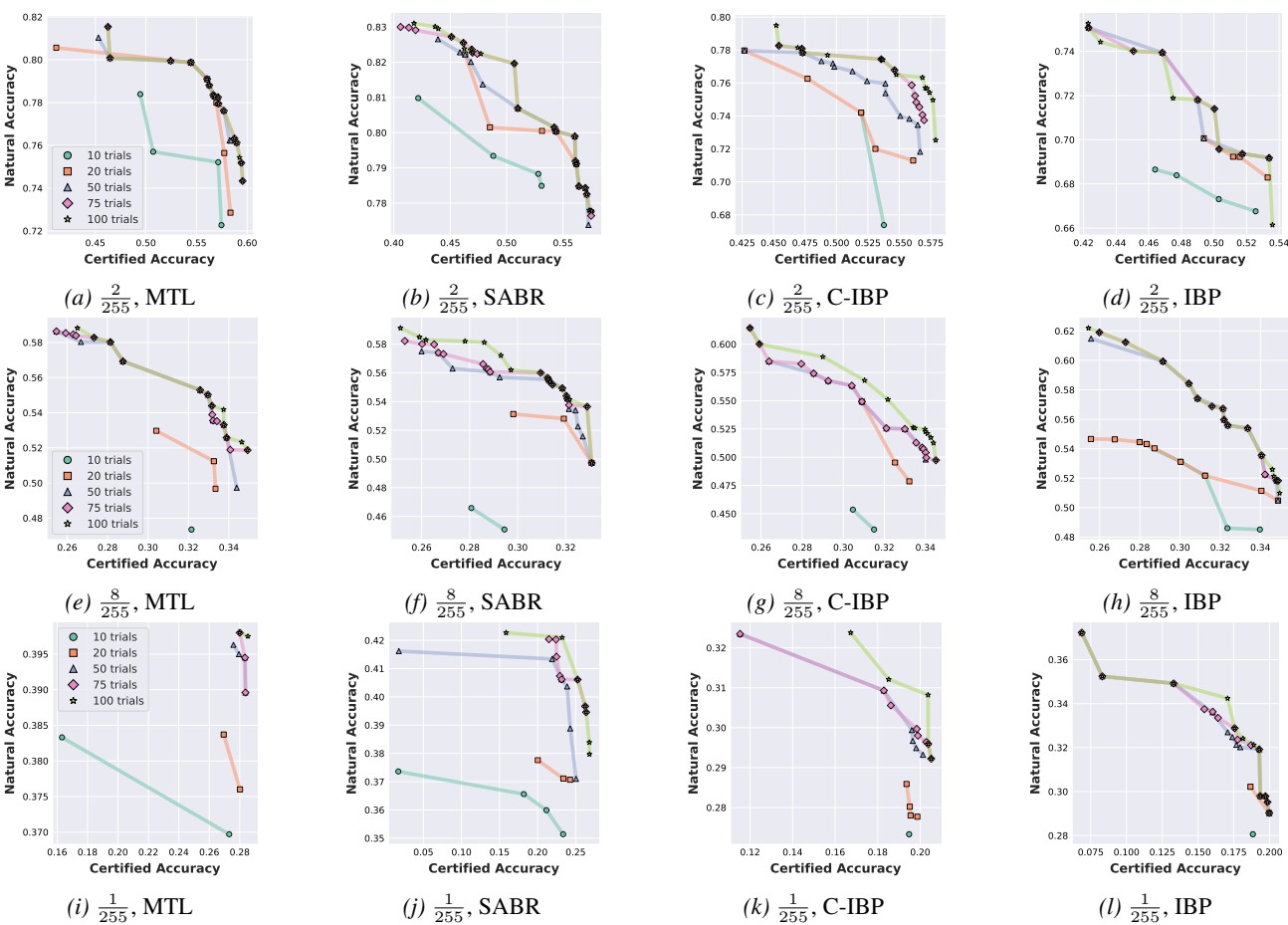

*Figure 12.* Evolution of the Pareto fronts of MTL-IBP, SABR, CROWN-IBP and IBP with varying evaluation budgets for hyperparameter optimisation given in trials per seed. It becomes apparent, that often 50 trials per seed are sufficient to approximate the Pareto front appropriately. Nevertheless, the optimisation algorithm continues to improve upon previous results until the budget is exhausted.

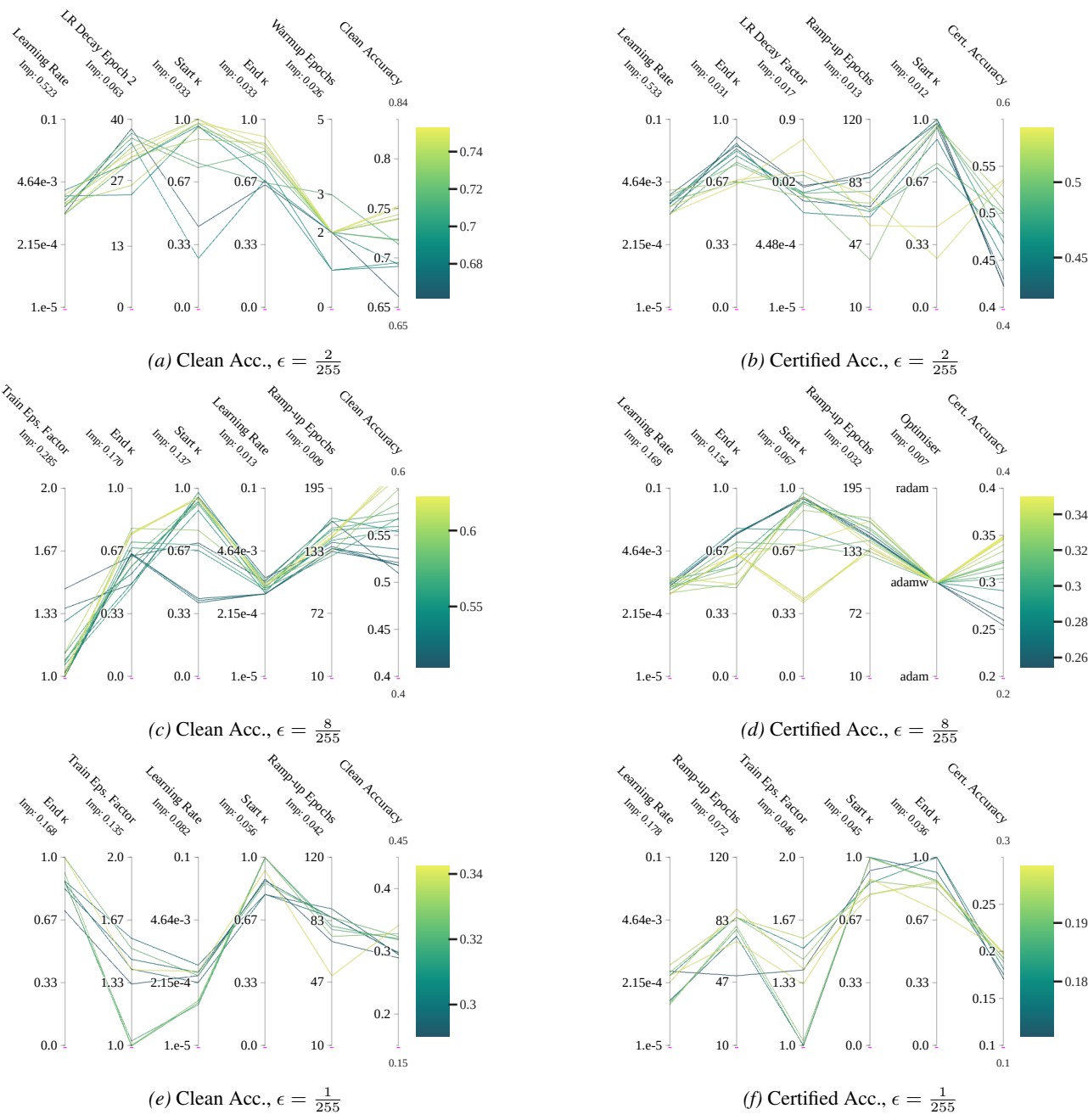

*Figure 13.* Parallel coordinates plot for the hyperparameter optimisation of IBP on CIFAR-10 ((a)-(d)) and Tiny ImageNet ((e)-(f)). In each plot, we show the five most important parameters with their importance scores for one of the two objectives along with the parameter values of configurations in the Pareto set.

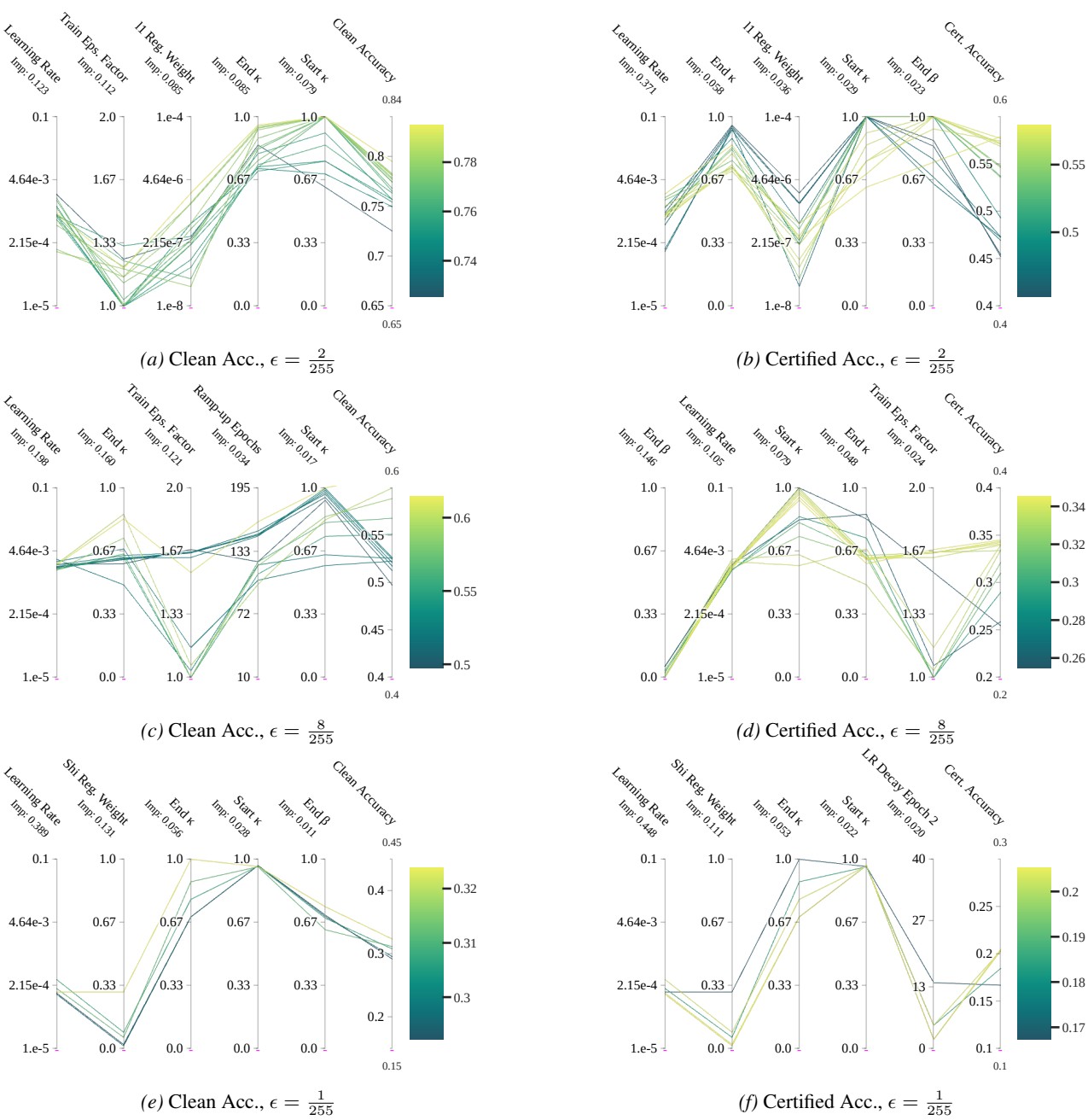

*Figure 14.* Parallel coordinates plot for the hyperparameter optimisation of CROWN-IBP on CIFAR-10 ((a)-(d)) and Tiny ImageNet ((e)-(f)). In each plot, we show the five most important parameters with their importance scores for one of the two objectives along with the parameter values of configurations in the Pareto set.

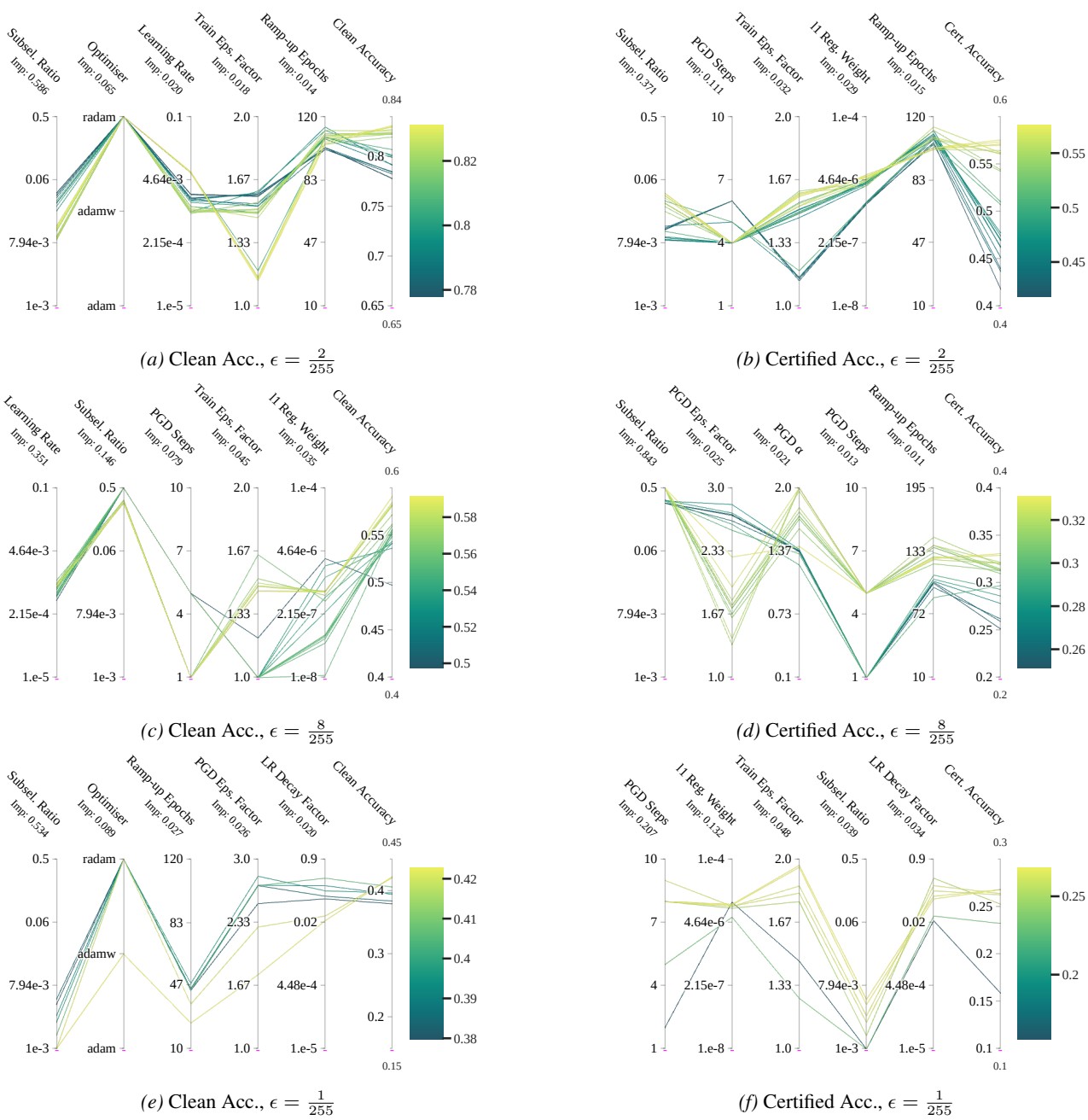

*Figure 15.* Parallel coordinates plot for the hyperparameter optimisation of SABR on CIFAR-10 ((a)-(d)) and Tiny ImageNet ((e)-(f)). In each plot, we show the five most important parameters with their importance scores for one of the two objectives along with the parameter values of configurations in the Pareto set.

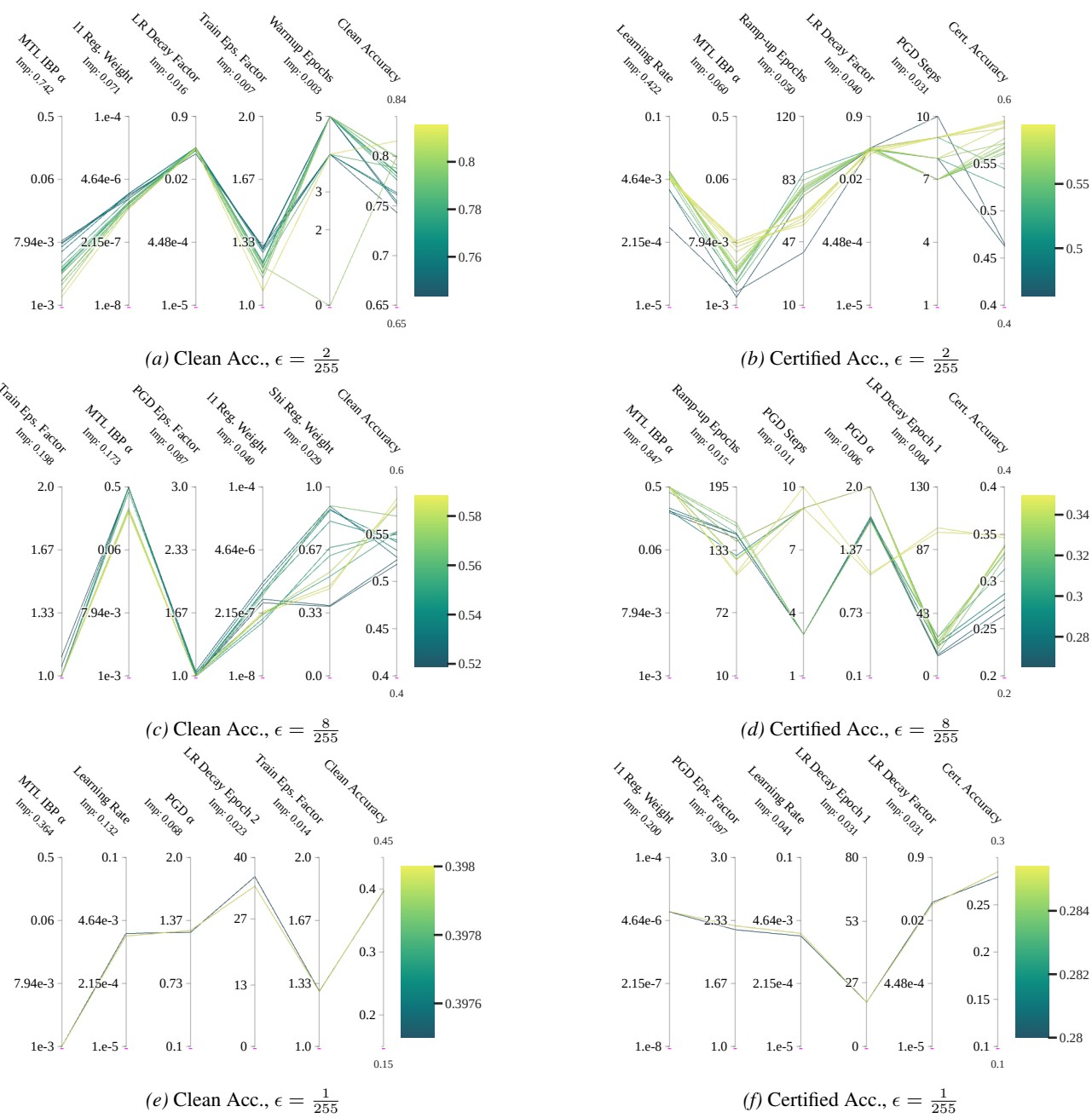

*Figure 16.* Parallel coordinates plot for the hyperparameter optimisation of MTL-IBP on CIFAR-10 ((a)-(d)) and Tiny ImageNet ((e)-(f)). In each plot, we show the five most important parameters with their importance scores for one of the two objectives along with the parameter values of configurations in the Pareto set.

*Table 9.* Overview of hyperparameter configurations found by our optimisation procedure on CIFAR10 $\epsilon = \frac{2}{255}$. We show the two configurations on the edges of the front, as well as one configuration from the middle.

*(a)* MTL-IBP

| Hyperparameter | (81.54, 62.82) | (79.87, 64.54) | (79.10, 64.61) |
|---|---|---|---|
| l1 Reg. Weight | 1.1e-6 | 1.1e-6 | 1.4e-6 |
| Learning Rate | 0.003 | 0.005 | 0.006 |
| LR Decay Epoch 1 | 15 | 11 | 12 |
| LR Decay Epoch 2 | 33 | 31 | 29 |
| LR Decay Factor | 0.128 | 0.135 | 0.115 |
| MTL IBP $\alpha$ | 0.001 | 0.002 | 0.002 |
| PGD Eps. Factor | 1.943 | 2.278 | 2.274 |
| PGD $\alpha$ | 1.712 | 1.633 | 1.635 |
| PGD Steps | 8 | 9 | 7 |
| Optimiser | RAdam | RAdam | RAdam |
| Ramp-up Epochs | 74 | 76 | 83 |
| Shi Reg. Weight | 0.356 | 0.367 | 0.414 |
| Train Eps. Factor | 1.079 | 1.174 | 1.168 |
| Warmup Epochs | 4 | 5 | 5 |

*(b)* SABR

| Hyperparameter | (83.10, 60.03) | (81.96, 64.11) | (79.90, 64.49) |
|---|---|---|---|
| l1 Reg. Weight | 1.4e-6 | 4.8e-6 | 4.4e-6 |
| Learning Rate | 0.007 | 9.1e-4 | 0.002 |
| LR Decay Epoch 1 | 25 | 27 | 31 |
| LR Decay Epoch 2 | 23 | 13 | 12 |
| LR Decay Factor | 0.088 | 0.124 | 0.114 |
| Optimiser | RAdam | RAdam | RAdam |
| Ramp-up Epochs | 105 | 104 | 108 |
| PGD $\alpha$ | 0.984 | 1.186 | 1.201 |
| PGD Eps. Factor | 2.64 | 2.85 | 2.934 |
| PGD Steps | 6 | 4 | 4 |
| Subsel. Ratio | 0.012 | 0.012 | 0.029 |
| Shi Reg. Weight | 0.896 | 0.389 | 0.394 |
| Train Eps. Factor | 1.144 | 1.547 | 1.527 |
| Warmup Epochs | 1 | 3 | 3 |

*(c)* IBP

| Hyperparameter | (75.26, 54.29) | (71.40, 55.54) | (69.18, 55.99) |
|---|---|---|---|
| l1 Reg. Weight | 1.8e-7 | 2.5e-8 | 8.7e-7 |
| Learning Rate | 9.7e-4 | 0.002 | 0.001 |
| LR Decay Epoch 1 | 19 | 36 | 43 |
| LR Decay Epoch 2 | 32 | 37 | 35 |
| LR Decay Factor | 0.015 | 0.031 | 0.038 |
| Optimiser | RAdam | RAdam | AdamW |
| Ramp-up Epochs | 89 | 67 | 75 |
| End $\kappa$ | 0.908 | 0.664 | 0.677 |
| Start $\kappa$ | 0.98 | 0.767 | 0.262 |
| Shi Reg. Weight | 0.733 | 0.735 | 0.244 |
| Train Eps. Factor | 1.248 | 1.025 | 1 |
| Warmup Epochs | 2 | 3 | 1 |

*(d)* CROWN-IBP

| Hyperparameter | (79.49, 54.13) | (77.69, 57.11) | (76.33, 60.94) |
|---|---|---|---|
| End $\beta$ | 0.874 | 1 | 1 |
| End $\kappa$ | 0.955 | 0.885 | 0.746 |
| Start $\kappa$ | 1 | 1 | 0.85 |
| l1 Reg. Weight | 2.4e-6 | 2.0e-7 | 6.6e-8 |
| Learning Rate | 9.3e-4 | 0.001 | 6.9e-4 |
| LR Decay Epoch 1 | 59 | 56 | 52 |
| LR Decay Epoch 2 | 5 | 10 | 0 |
| LR Decay Factor | 0.001 | 0.002 | 0.013 |
| Optimiser | Adam | Adam | AdamW |
| Ramp-up Epochs | 54 | 54 | 74 |
| Shi Reg. Weight | 0 | 0.191 | 0.128 |
| Train Eps. Factor | 1.2 | 1.009 | 1 |
| Warmup Epochs | 3 | 4 | 3 |

*Table 10.* Overview of hyperparameter configurations found by our optimisation procedure on CIFAR10 $\epsilon = \frac{8}{255}$. We show the two configurations on the edges of the front, as well as one configuration from the middle.

*(a)* MTL-IBP

| Hyperparameter | (58.82, 32.55) | (55.03, 34.72) | (51.87, 35.74) |
|---|---|---|---|
| l1 Reg. Weight | 2.1e-7 | 1.4e-7 | 3.6e-7 |
| Learning Rate | 0.002 | 0.002 | 0.001 |
| LR Decay Epoch 1 | 14 | 27 | 99 |
| LR Decay Epoch 2 | 43 | 49 | 59 |
| LR Decay Factor | 0.106 | 0.151 | 0.012 |
| MTL IBP $\alpha$ | 0.216 | 0.5 | 0.488 |
| PGD Eps. Factor | 1 | 1 | 1 |
| PGD $\alpha$ | 1.68 | 1.661 | 1.137 |
| PGD Steps | 3 | 3 | 10 |
| Optimiser | RAdam | RAdam | AdamW |
| Ramp-up Epochs | 145 | 160 | 111 |
| Shi Reg. Weight | 0.465 | 0.681 | 0.37 |
| Train Eps. Factor | 1 | 1 | 1.102 |
| Warmup Epochs | 0 | 0 | 4 |

*(b)* SABR

| Hyperparameter | (59.12, 28.68) | (58.12, 31.55) | (54.93, 34.96) |
|---|---|---|---|
| l1 Reg. Weight | 6.4e-7 | 6.6e-7 | 7.9e-8 |
| Learning Rate | 8.2e-4 | 9.1e-4 | 5.0e-4 |
| LR Decay Epoch 1 | 95 | 91 | 94 |
| LR Decay Epoch 2 | 0 | 0 | 39 |
| LR Decay Factor | 0.023 | 0.018 | 0.209 |
| Optimiser | AdamW | AdamW | RAdam |
| Ramp-up Epochs | 102 | 106 | 147 |
| PGD $\alpha$ | 1.362 | 1.373 | 2 |
| PGD Eps. Factor | 2.71 | 2.736 | 1.345 |
| PGD Steps | 1 | 1 | 5 |
| Subsel. Ratio | 0.305 | 0.336 | 0.5 |
| Shi Reg. Weight | 0.378 | 0.407 | 0.769 |
| Train Eps. Factor | 1.457 | 1.481 | 1 |
| Warmup Epochs | 4 | 4 | 5 |

*(c)* IBP

| Hyperparameter | (62.20, 28.33) | (56.87, 32.83) | (50.98, 35.35) |
|---|---|---|---|
| l1 Reg. Weight | 8.6e-8 | 2.0e-7 | 9.5e-8 |
| Learning Rate | 8.5e-4 | 7.6e-4 | 0.001 |
| LR Decay Epoch 1 | 53 | 65 | 48 |
| LR Decay Epoch 2 | 33 | 30 | 18 |
| LR Decay Factor | 0.136 | 0.148 | 0.065 |
| Optimiser | AdamW | AdamW | AdamW |
| Ramp-up Epochs | 148 | 129 | 163 |
| End $\kappa$ | 0.759 | 0.715 | 0.641 |
| Start $\kappa$ | 0.955 | 0.699 | 0.711 |
| Shi Reg. Weight | 0.427 | 0.13 | 0.429 |
| Train Eps. Factor | 1.008 | 1 | 1.467 |
| Warmup Epochs | 3 | 2 | 4 |

*(d)* CROWN-IBP

| Hyperparameter | (61.43, 28.72) | (55.11, 33.77) | (49.73, 34.78) |
|---|---|---|---|
| End $\beta$ | 0.058 | 0.009 | 0 |
| End $\kappa$ | 0.837 | 0.634 | 0.601 |
| Start $\kappa$ | 1 | 0.745 | 0.936 |
| l1 Reg. Weight | 2.7e-7 | 8.1e-8 | 3.4e-7 |
| Learning Rate | 0.002 | 0.002 | 0.002 |
| LR Decay Epoch 1 | 50 | 68 | 47 |
| LR Decay Epoch 2 | 46 | 24 | 44 |
| LR Decay Factor | 0.111 | 0.002 | 0.064 |
| Optimiser | RAdam | Adam | RAdam |
| Ramp-up Epochs | 162 | 111 | 123 |
| Shi Reg. Weight | 0.293 | 0.606 | 0.215 |
| Train Eps. Factor | 1.554 | 1 | 1.675 |
| Warmup Epochs | 0 | 2 | 0 |

*Table 11.* Overview of hyperparameter configurations found by our optimisation procedure on TinyImageNet $\epsilon = \frac{1}{255}$. We show the two configurations on the edges of the front, as well as one configuration from the middle.

*(a)* MTL-IBP

| Hyperparameter | (39.80, 30.45) | (39.75, 30.67) |
|---|---|---|
| l1 Reg. Weight | 7.2e-6 | 7.3e-6 |
| Learning Rate | 0.002 | 0.003 |
| LR Decay Epoch 1 | 19 | 19 |
| LR Decay Epoch 2 | 34 | 36 |
| LR Decay Factor | 0.062 | 0.055 |
| MTL IBP $\alpha$ | 0.001 | 0.001 |
| PGD Eps. Factor | 2.24 | 2.282 |
| PGD $\alpha$ | 1.27 | 1.25 |
| PGD Steps | 7 | 7 |
| Optimiser | RAdam | RAdam |
| Ramp-up Epochs | 86 | 87 |
| Shi Reg. Weight | 0.306 | 0.321 |
| Train Eps. Factor | 1.294 | 1.295 |
| Warmup Epochs | 5 | 5 |

*(b)* SABR

| Hyperparameter | (42.27, 23.13) | (42.10, 27.42) | (40.61, 28.86) |
|---|---|---|---|
| l1 Reg. Weight | 1.2e-5 | 6.0e-6 | 9.3e-6 |
| Learning Rate | 4.3e-4 | 0.003 | 0.002 |
| LR Decay Epoch 1 | 60 | 51 | 62 |
| LR Decay Epoch 2 | 0 | 29 | 29 |
| LR Decay Factor | 0.022 | 0.03 | 0.289 |
| Optimiser | AdamW | RAdam | RAdam |
| Ramp-up Epochs | 25 | 36 | 44 |
| PGD $\alpha$ | 1.06 | 1.291 | 1.006 |
| PGD Eps. Factor | 1.783 | 2.285 | 2.721 |
| PGD Steps | 2 | 5 | 8 |
| Subsel. Ratio | 0.001 | 0.001 | 0.002 |
| Shi Reg. Weight | 0.429 | 0.801 | 0.824 |
| Train Eps. Factor | 1.463 | 1.267 | 1.777 |
| Warmup Epochs | 1 | 2 | 2 |

*(c)* IBP

| Hyperparameter | (34.24, 20.79) | (32.12, 21.60) | (31.92, 21.82) |
|---|---|---|---|
| l1 Reg. Weight | 5.8e-6 | 1.6e-6 | 1.3e-6 |
| Learning Rate | 3.9e-4 | 8.3e-5 | 7.5e-5 |
| LR Decay Epoch 1 | 43 | 28 | 31 |
| LR Decay Epoch 2 | 10 | 38 | 40 |
| LR Decay Factor | 0.042 | 9.5e-4 | 0.002 |
| Optimiser | Adam | RAdam | RAdam |
| Ramp-up Epochs | 51 | 78 | 80 |
| End $\kappa$ | 1 | 0.877 | 0.871 |
| Start $\kappa$ | 0.931 | 1 | 0.997 |
| Shi Reg. Weight | 0 | 0.653 | 0.622 |
| Train Eps. Factor | 1.403 | 1 | 1.027 |
| Warmup Epochs | 4 | 0 | 1 |

*(d)* CROWN-IBP

| Hyperparameter | (32.38, 20.71) | (31.21, 20.96) | (30.82, 22.20) |
|---|---|---|---|
| End $\beta$ | 0.75 | 0.629 | 0.69 |
| End $\kappa$ | 1 | 0.881 | 0.788 |
| Start $\kappa$ | 0.962 | 0.96 | 0.964 |
| l1 Reg. Weight | 4.3e-8 | 3.3e-6 | 5.8e-6 |
| Learning Rate | 1.6e-4 | 1.9e-4 | 2.9e-4 |
| LR Decay Epoch 1 | 46 | 52 | 46 |
| LR Decay Epoch 2 | 14 | 5 | 5 |
| LR Decay Factor | 4.9e-4 | 2.3e-4 | 3.3e-4 |
| Optimiser | Adam | Adam | Adam |
| Ramp-up Epochs | 65 | 63 | 59 |
| Shi Reg. Weight | 0.301 | 0.062 | 0.088 |
| Train Eps. Factor | 1.277 | 1.051 | 1.031 |
| Warmup Epochs | 4 | 3 | 3 |

# I. Pseudo-code

Algorithm 1 provides pseudo-code for our proposed constrained multi-objective hyperparameter optimisation method for certified training of deep neural networks. Line 1 gathers the initial random samples, which are evaluated in line 2. In line 4, we determine which configurations belong to the Pareto set. The optimisation loop then begins with fitting the surrogate models in line 6. Line 7 then optimises the acquisition function to decide on the next candidate configuration. We then evaluate the configuration and add it to the set of evaluated configurations in line 8. Lastly, in line 9, we determine which configurations belong to the Pareto set based on the updated set of evaluated configurations.

---

**Algorithm 1** Multi-objective hyperparameter optimisation for certified training

---

1: **Input:** total budget $b$, initial sample size $r$, certified training method $t$, incomplete verification method $v$, dataset $D$, min. clean acc. constraint $c_{\text{clean}}$, min. cert acc. constraint $c_{\text{cert}}$
2: Initialise $\zeta$ with $r$ randomly sampled points
3: $\zeta \leftarrow \{(\lambda, v(t(D, \lambda))) \mid \lambda \in \zeta\}$
4: $P \leftarrow \{(\lambda, (m_{\text{clean}}, m_{\text{cert}})) | \nexists_{(\lambda', (m'_{\text{clean}}, m'_{\text{cert}})) \in \zeta} (m_{\text{clean}}, m_{\text{cert}}) \prec (m'_{\text{clean}}, m'_{\text{cert}}) \land (m_{\text{clean}} \geq c_{\text{clean}} \land m_{\text{cert}} \geq c_{\text{cert}})\}$
5: **while** budget $b$ is not exhausted **do**
6: $\quad S_{\text{clean}}, S_{\text{cert}}, S_{\text{clean cond}}, S_{\text{cert cond}} \leftarrow \text{fit}(\zeta)$
7: $\quad \lambda_t \leftarrow \arg\max \text{EHVI}(S_{\text{clean}}, S_{\text{cert}}, S_{\text{clean cond}}, S_{\text{cert cond}}, P, c_{\text{clean}}, c_{\text{cert}})$
8: $\quad \zeta \leftarrow \zeta \cup \{(\lambda, v(t(D, \lambda)))\}$
9: $\quad P \leftarrow \{(\lambda, (m_{\text{clean}}, m_{\text{cert}})) | \nexists_{(\lambda', (m'_{\text{clean}}, m'_{\text{cert}})) \in \zeta} (m_{\text{clean}}, m_{\text{cert}}) \prec (m'_{\text{clean}}, m'_{\text{cert}}) \land (m_{\text{clean}} \geq c_{\text{clean}} \land m_{\text{cert}} \geq c_{\text{cert}})\}$
10: **end while**
11: **Return** $P$

---

