# OpenReview forum: "Rethinking Evaluation Paradigms in IBP-based Certified Training"
_ICML.cc/2026/Conference — ICML 2026 regular_

### Official Review · Reviewer_6RwH · 2026-03-01

**Soundness:** 3
**Presentation:** 4
**Significance:** 1
**Originality:** 1
**Overall Recommendation:** 3
**Confidence:** 5

**Summary:**

The paper proposes to rethink the evaluation paradigm of algorithms that train neural networks for certified robustness to adversarial perturbations. Specifically, they propose to switch from single trade-off to reporting a Pareto curve of trade-offs between standard and certified accuracy. In addition, the paper proposes a unified evaluation procedure for existing certified training algorithms based on hyper-parameter optimization via Bayesian optimization. Conclusions pertaining to the systematic evaluation are reported.

**Compliance With Llm Reviewing Policy:**

Affirmed.

**Final Justification:**

While the study is interesting, I think its weaknesses, further confirmed after the authors' responses, outweigh its strengths. I hence retain my original score.
In particular:
- I believe the cost associated with the procedure is too large and prevents systematic adoption. Given that progress in the area has relatively stalled, I am not sure that increasing the computational cost underlying new algorithms is the preferable way forward;
- It is desirable to provide Pareto curves corresponding to the algorithms' main hyper-parameter (e.g. $\alpha$ for MTL-IBP) and compare it with the full HPO;
- I think it's important to carry out a reduced HPO study directly using complete verification performance under a reasonable timeout (at least 300s).
- I am still unconvinced about the surprisingly small Pareto curve on TinyImageNet: this could be linked with the use of incomplete verifiers or to other potential methodological limitations;
- Trade-off differences between seemingly similar loss functions are not explained but merely observed: given the similar technical properties of SABR and MTL-IBP (and of the other losses presented in the MTL-IBP work), Pareto curve differences may be linked to the employed HPO procedure as opposed to the methods themselves. The work cannot disambiguate on this.

**Key Questions For Authors:**

- Verification is carried out with incomplete verifiers during the hyper-parameter optimization procedure. However, it is very well known in the area (see Figure 7 in SABR, Figure 1 in MTL-IBP) that maximizing certified accuracy with an incomplete verifier will not lead to the same hyper-parameters as aiming directly at BaB accuracy. This could potentially bias the entire tuning process. Could the authors provide results on directly tuning on BaB accuracy on a subset of the experiments?
- The MTL-IBP paper introduces two similar methods (CC-IBP and Exp-IBP), which however may display different Pareto curves under the proposed HPO procedure, potentially as different as SABR from MTL-IBP. Could the authors provide these on at least CIFAR-10 2/255? Why did you focus on MTL-IBP specifically?
- I feel like it would be reasonable to provide the actual list of hyper-parameters associated to the central few points in the Pareto curves.
- Why is the SABR and MTL-IBP Pareto curve range so limited on TinyImageNet? Intuitively, tuning $\tau$ or $\alpha$ alone should yield a wide range of trade-offs. These results feel somewhat incomplete.

**Limitations:**

Yes

**Strengths And Weaknesses:**

**Strengths.**

- The high-level point being made (switching to Pareto front evaluation) is definitely a valid and interesting one. So far, certified training algorithms have focused on strictly improving on the prevalent trade-off between standard and certified accuracy, but this does not of course convey the full picture of an algorithm's performance.
- The outcome of the tuning process shows interesting conclusions (IBP performs better than more recent methods in large-perturbations in some regimes) and reports strictly-better trade-offs than those found in the existing literature.
- The paper is very well presented.

**Weaknesses.**

- In spite of the above strengths, I believe the significance of the presented idea is not necessarily large. It is quite well known in the community that algorithmic improvements have somewhat stalled in recent years, and single trade-off reporting is mostly due to the overhead associated to deriving the Pareto front. Considering the cost associated with certified training (at training and at verification), it is quite utopian, I would say, to expect that new algorithms would adopt this evaluation approach.
- The technical contribution on its own is of course quite minor, as it is mostly linked to the use of existing hyper-parameter optimization routines.
- This is not the first systematic evaluation of certified training algorithms, as acknowledged by the authors. While the conclusions do differ a bit, they are still relatively in line with CTBench.
- I did not find the evaluation to be particularly inclusive (see questions).

Overall, I do not feel like this work meets the standard acceptance criteria for ICML papers, and would perhaps be more suitable for different venues in the ML community.

---

> ### Author Rebuttal · Authors · 2026-03-31
>
> We sincerely thank the reviewer for the careful evaluation. We are encouraged that our Pareto-based evaluation is recognized as “valid and interesting,” with “interesting conclusions” and “strictly better trade-offs than those found in the literature.” Below we address the reviewer’s questions and suggestions.
>
> **Q: Adaptation of the proposed evaluation approach is “utopian” due to computational costs.**
>
> While evaluating certified training is computationally costly, we respectfully disagree that our method is “utopian.” As Sec. 3 shows, the number of evaluations is similar to related work (around 300). App. F shows that Pareto fronts can be recovered with shorter verification timeouts, and good approximations are achieved after 150 evaluations. In addition, future work can build on our results without re-running the full evaluation, but running the evaluation procedure only for any newly introduced method.
>
> We respectfully disagree that the “significance of the presented idea is not necessarily large.” Our paradigm provides a nuanced view for the performance of the methods by providing, for the first time, full Pareto fronts. This allows new methods to report improvements in specific trade-off regions that are as relevant to practitioners and researchers as single, arbitrarily chosen points commonly reported today.
>
> **Q: Minor technical contribution.**
>
> While HPO using Bayesian opt. and IBP-based certified training are established, their combination proved non-trivial. Enabling our evaluation required a proxy objective based on incomplete verification, constrained opt., and carefully designed search spaces. Our primary contribution is conceptual: a novel evaluation paradigm for certified training, with a simple and efficient implementation. ICML explicitly lists evaluation as a topic of interest in the call for papers.
>
> **Q: Not the first systematic evaluation of IBP-based certified training.**
>
> CTBench attempted systematic evaluation, but we respectfully disagree that our conclusions are “relatively in line” with it. Sec. 5 shows that CTBench tuned methods towards certified accuracy, reducing clean accuracy. Our approach achieves similar certified accuracy at substantially higher clean accuracy. Our multi-obj. opt. enables comparisons across the entire trade-off space. For example, at larger perturbation radii, IBP performs best at higher natural accuracies, whereas MTL-IBP/SABR excel at higher certified accuracies, one of our novel conclusions not revealed by CTBench.
>
> **Q: Results on directly tuning on BaB accuracy are missing.**
>
> Directly tuning for BaB accuracy is computationally infeasible, as even with timeouts of 60s per instance , for only 8 networks trained and evaluated after more than 4 days, while the optimisations that use incomplete verification are already done within the same time budget. The results from this procedure are inferior to the results obtained by using incomplete verification. Moreover, Fig. 1 (MTL-IBP) and Fig. 7 (SABR) show that configurations with the highest BaB accuracy lie on the Pareto front of natural _vs_ incomplete verification accuracy, which are identified by our procedure and included in the final evaluation.
>
> **Q: Evaluation on CC-IBP and EXP-IBP is missing.**
>
> We focused on MTL-IBP as the main representative of Expressive Losses. Experiments on CC-IBP and EXP-IBP are ongoing and will be included in the camera-ready version. Preliminary results on CIFAR10 with ε=2/255 show for CC-IBP using incomplete verification configurations:
>
> (.8122, .438)\
> (.7783, .567)\
> (.7939, .544)\
> (.790, 0551)\
> (.8075, .510)\
> (.7922, .549)\
> (.7977, .538)\
> (.8231, .436)\
> (.7627, .581)\
> (.7589, .582)\
> (.7647, .580)\
> (.7578, .587)\
> (.7513, .588)\
> (.8245, .415)\
> (.7487, .590)
>
> For EXP-IBP, we have the following configurations in the Pareto front:
>
> (.8007, .467)\
> (.8017, .461)\
> (.8095, .447)\
> (.7155, .561)\
> (.7983, .477)\
> (.7476, .559)\
> (.8023, .451)\
> (.7549, .532)\
> (.8208, .418)\
> (.7519, .551)\
> (.7925, .507)\
> (.7831, .532)\
> (.7104, .563)
>
> Here, the first and second numbers show natural and certified accuracy, respectively. We plan to run complete verification on these methods and evaluate them on other datasets and epsilons.
>
> **Q: List of hyperparameters for central Pareto points is missing.**We will include a detailed overview.
>
> **Q: Why is the range of the Pareto curve limited on TinyImageNet?**
>
> We agree that this result might be unexpected, but consider this an important empirical finding. In higher-dimensional settings, like TinyImageNet, IBP-based training does not achieve Pareto-optimal performance across the full trade-off space, highlighting the need for new methods in underperforming regions.
>
> Lastly, we reiterate our gratitude for the thorough review. Having addressed the potential weaknesses, we are confident that our novel Pareto-based evaluation paradigm for certified training provides richer insights and more nuanced comparisons, revealing previously unobserved trade-offs.

---

> > ### Author Rebuttal · Reviewer_6RwH · 2026-04-01
> >
> > I sincerely thank the reviewers for their response.
> > However, I do not think my main points were successfully addressed.
> >
> > I still believe that the entire procedure is way too costly to be adopted by new certified training schemes (the fact that CTBench is costly too is not of great advantage: to the best of my knowledge, it has not been adopted by the literature in the area).
> > And I disagree that the best trade-offs between standard accuracy and incomplete-based certified accuracy match those using complete verification. While the granularity of the SABR and MTL-IBP experiments on this from the original papers is perhaps too large, Figure 1(c) from the MTL-IBP paper does show quite clearly that the trade-offs do not necessarily match.
> > Providing experiments using complete verification on at least one method/setup is, I believe, necessary.
> > I would really believe this mismatch can be exposed by repeating the sensitivity analysis at a finer granularity: this would be perhaps a good addition to the paper, in any case.
> > I also have the tendency to speculate that this may be perhaps linked to the different Pareto curve seen on TinyImageNet: I wonder whether the use of a complete verifier would change things.
> >
> > Overall, I can see the merits of the work, as listed in my original review. I just believe that overall, it is below the ICML acceptance threshold.

---

> > > ### Author Response · Authors · 2026-04-06
> > >
> > > We thank the reviewer for the detailed and thoughtful feedback. We regret that we were not yet successful in addressing the key concerns of the reviewer. In the following we give further clarification on our methodology and the scalability concerns.
> > >
> > > We fully agree that Pareto fronts obtained with incomplete and complete verification do not generally coincide, as also observed in prior work (e.g., SABR, MTL-IBP). Importantly, our approach does not rely on such a match.
> > >
> > > First, during multi-objective HPO, we optimise over natural accuracy and incomplete certified accuracy. This stage can be seen as a search mechanism to identify promising configurations. Importantly, the referenced figures in SABR and MTL-IBP show that the configurations that lie on the Pareto front of natural vs. certified accuracy using CROWN contain the best-performing configurations under complete verification, which aligns with our results. Thus, discovering Pareto-optimal configurations under the proxy objective also discovers well-performing configurations under complete verification. Second, after optimisation, we perform complete verification on all configurations on the proxy Pareto front (after clustering near-duplicates). Thus, we do not assume alignment between proxy and complete trade-offs; we only require that the proxy identifies a sufficiently rich candidate set. Empirically, we find that there is strong Spearman rank correlation between proxy and complete-certified accuracy (Appendix D.4), indicating that the proxy is a useful ranking signal.
> > >
> > > While optimising directly for BaB-certified accuracy would be ideal in principle, it is currently infeasible in practice. Even with an aggressive per-instance verification timeout of 60 seconds, only 8 models were evaluated after more than 4 days of running time, which resulted in substantially worse results than our proxy based approach.
> > >
> > > Crucially, the fact that we observed consistent improvements over prior work (e.g., SABR, MTL-IBP, CTBench) in terms of BaB-certified trade-offs under comparable or smaller budgets strongly suggests that the proxy is sufficiently aligned to guide HPO towards well-performing configurations under complete verification, even though the proxy and BaB trade-offs do not coincide pointwise. We will make this two-stage structure and its implications more explicit in Sec. 4 and strengthen references to Appendix D.4.
> > >
> > > We thank the reviewer for suggesting a simple α-sweep baseline for MTL-IBP. We agree this is a valuable comparison and will include it. Specifically, we will extend the α-sensitivity analysis for CIFAR-10 (ε = 2/255), report both incomplete and complete-certified accuracy, and directly compare the resulting curve to the Pareto front from our full protocol. Preliminary results indicate that our method strictly dominates these one-dimensional sweeps demonstrating the benefit of full multi-objective tuning.
> > >
> > > Regarding computational costs and practical adoption, Appendix F shows that high-quality Pareto fronts can be obtained with significantly reduced verification timeouts and roughly half the HPO budget, with only minor degradation. In practice, researchers do not need to recompute all baselines since existing Pareto fronts can be reused, and only new methods need to be evaluated. Combined with cheaper approximations, this makes the approach considerably more practical than it may initially appear.
> > >
> > > In summary, (i) we are aware that incomplete verification trade-offs do not necessarily coincide with complete verification, (ii) we use incomplete verification only to discover a rich candidate set, followed by full αβ-CROWN evaluation, (iii) this strategy empirically improves over prior BaB-based results, and (iv) direct optimisation on BaB remains infeasible, making a well-chosen proxy necessary.
> > >
> > > We thank the reviewer again for the insightful comments and believe the additional clarifications and experiments will further strengthen the paper.

---

### Official Review · Reviewer_yj1j · 2026-03-07

**Soundness:** 4
**Presentation:** 3
**Significance:** 4
**Originality:** 4
**Overall Recommendation:** 6
**Confidence:** 4

**Summary:**

Training neural networks requires carefully tuning hyperparameters, but previous works on training provably robust neural networks seem to have disregarded this aspect too much. This paper applies automatic hyperparameter tuning to reveal the true best performance existing provably robust training approaches can provide. The paper places special emphasis on the Pareto front between natural accuracy and provably robust accuracy.

**Compliance With Llm Reviewing Policy:**

Affirmed.

**Final Justification:**

The rebuttal has resolved the minor concerns I had in my review. Furthermore, the authors agreed with my suggestion to prominently report results trained on a validation set separate from the test set, thereby revealing that the community's current evaluation practice inflates results, as already apparent from the preliminary results provided during the rebuttal period.

This paper is a unique opportunity to encourage better evaluation practice in the certified training community. Coupled with the paper's message that proper hyperparameter tuning yields improvements comparable to the recent technical advancements in certified training, this paper can have an *exceptional* impact on the evaluation culture in certified training and beyond. For this reason, I updated my review to now **strongly recommend accepting** this paper.

**Key Questions For Authors:**

1. You tune the hyperparameter on the test set, as is common practice in this community. However, it is a poor practice overall. To establish better practice, it would be good to provide results tuned on a separate evaluation set as well (or better: primarily). Do you have such results?

**Typos and such**
- "revealing previously unreported complementary performance (of ...)" sounds better than "performance complementarities". (line 40)
- Not all attacks depend on local gradient information (line 43).
- Line 72 states that the robustness-accuracy tradeoff is **mostly** underexplored for deterministic training methods. Please add the corresponding citations for methods that explore the tradeoff or delete the "mostly*.
- There are some double periods in the paper..
- It would be good to introduce the notation used in lines 112-115.
- Line 377: missing "the".
- Capitalisation in the References Section. E.g. line 465: **M**onte-**C**arlo **B**ayesian.

**Limitations:**

Yes

**Strengths And Weaknesses:**

Major points are highlighted in **bold**.

- **Significant contribution: evaluation using properly tuned hyperparameters is key for advancing machine learning.**
- Overall good presentation with minor issues.
- The paper poses the blanket statement that natural accuracy and provably robust accuracy are inherently conflicting. This phenomenon clearly appears in practice, but should at least be substantiated with references in the submission. I'm not happy with how universally this statement is posed in the introduction, but this does not affect the soundness or significance of the results.
- Missing citations for multi-objective hyperparameter optimisation, multi-objective Bayesian optimisation, and the claim that l1 reg has proven good for training in lines 171, 185, and 207.
- **Claim "since the optimisation objectives are independent from each other, we model them using distinct Gaussian processes" contradicts the earlier statement that natural and provably robust accuracy conflict (lines 20-21 and 190-191).**
- The evaluation in Appendix D.4 of how good the robustness underapproximations predict the provably robust accuracy is crucial. It should be linked in lines 262-269. Ideally, I think it should go into the main body.
- **Strong evaluation with standard architectures and perturbation radius.**

There is always some room to argue with hyperparameter optimisation whether all relevant hyperparameters were considered. For example, the paper does not tune loss fusion yes/no (line 292). However, in my opinion, the paper does a good job of selecting the key hyperparameters. It's results are highly valuable for the robust deep learning community.

---

> ### Author Rebuttal · Authors · 2026-03-31
>
> We sincerely thank the reviewer for the careful and constructive evaluation. We are particularly encouraged that the reviewer recognises the paper as a “significant contribution” and highlights that “evaluation using properly tuned hyperparameters is key for advancing machine learning.” We are also grateful for the strong overall assessment (excellent soundness, significance, and originality) and the recognition that our work provides “highly valuable” results for the robust deep learning community. Below we address the reviewer’s questions and suggestions.
>
> **Q: Gaussian process modelling and objective independence.**
>
> The reviewer notes a potential contradiction between modelling objectives with separate Gaussian processes and the earlier statement that natural and provably verified accuracy may conflict. Our use of distinct Gaussian processes follows standard practice in multi-objective Bayesian optimisation \[1]. It does not imply that the objectives themselves are independent in practice, but is required when it is impossible to mathematically infer the value of one objective from the other. We will clarify this point in the text to avoid confusion.
>
> **Q: Tuning on the test set.**
>
> We agree with the reviewer that tuning on the test set is “a poor practice overall,” even though it remains common practice in the certified training literature. Our goal was to ensure comparability with prior work, which almost exclusively reports results obtained in this way. At the same time, we agree that establishing better practices is important. We therefore conducted preliminary experiments using a separate evaluation set for hyperparameter tuning and observed very similar Pareto fronts and relative method rankings with, at the same time, markedly reduced absolute performance. We will include these additional results in the Appendix of the camera-ready version. Preliminary results after the full optimisation procedure have models with the following performance in the Pareto fronts (first number is for natural accuracy, second is for certified accuracy using incomplete verification):
>
> MTL-IBP
>
> (0.7708, 0.535)
>
> (0.7694, 0.544)
>
> (0.7613, 0.563)
>
> (0.7852, 0.530)
>
> (0.7856, 0.512)
>
> (0.7964, 0.450)
>
> (0.7509, 0.569)
>
> (0.7346, 0.575)
>
> (0.8035, 0.460)
>
> (0.7291, 0.579)
>
> SABR
>
> (0.8168, 0.457)
>
> (0.8157, 0.449)
>
> (0.7974, 0.491)
>
> (0.7887, 0.526)
>
> (0.8033, 0.483)
>
> (0.7701, 0.556)
>
> (0.7644, 0.564)
>
> (0.8048, 0.477)
>
> (0.7946, 0.490)
>
> (0.7776, 0.541)
>
> (0.7692, 0.553)
>
> (0.7913, 0.515)
>
> (0.7551, 0.563)
>
> (0.7509, 0.564)
>
> **Q: Choice of tuned hyperparameters.**
>
> As the reviewer notes, “there is always some room to argue with hyperparameter optimisation whether all relevant hyperparameters were considered.” Our goal was to focus on the most influential hyperparameters identified in prior work while keeping the optimisation problem tractable. We appreciate the reviewer’s assessment that “the paper does a good job of selecting the key hyperparameters” and that the resulting analysis is “highly valuable for the robust deep learning community.”
>
> **Q: Minor comments and typos.**
>
> We thank the reviewer for the detailed suggestions. We will incorporate the proposed wording improvements, clarify the statement regarding attacks that rely on gradient information, add references on the robustness–accuracy trade-off, and introduce the notation of accessing vector entries with brackets used in lines 112–115. We will also fix the typographical issues noted in the review. Additionally, we will include the missing citations for multi-objective hyperparameter optimisation, multi-objective Bayesian optimisation, and L1 regularisation, and add references supporting the statement that natural accuracy and provably robust accuracy are inherently conflicting in the introduction.
>
> We thank the reviewer again for the thoughtful feedback and positive assessment. We believe that our Pareto-based evaluation and systematic hyperparameter tuning provide an important step toward more reliable and informative evaluation of certified training methods, and strongly believe that the suggested revisions will further strengthen the paper.
>
> \[1] Daulton et al., Differentiable Expected Hypervolume Improvement for Parallel Multi-Objective Bayesian Optimization, NeurIPS 2020

---

> > ### Author Rebuttal · Reviewer_yj1j · 2026-04-01
> >
> > My concerns were mostly presentation issues that the authors have acknowledged. I trust they will address them adequately in the camera-ready. For the most impactful presentation issue I raised:
> >
> > > Claim "since the optimisation objectives are independent from each other, we model them using distinct Gaussian processes" contradicts the earlier statement that natural and provably robust accuracy conflict (lines 20-21 and 190-191).
> >
> > the authors have acknowledged that this was phrased incorrectly in the initial submission. In their rebuttal, they correctly wrote that assuming independence is technically necessary. I agree and am satisfied with this.
> >
> > > We agree with the reviewer that tuning on the test set is “a poor practice overall,” even though it remains common practice in the certified training literature. Our goal was to ensure comparability with prior work, which almost exclusively reports results obtained in this way. At the same time, we agree that establishing better practices is important. We therefore conducted preliminary experiments using a separate evaluation set for hyperparameter tuning and observed very similar Pareto fronts and relative method rankings with, at the same time, markedly reduced absolute performance. We will include these additional results in the Appendix of the camera-ready version.
> >
> > The comparability to existing work is a valid concern and is necessary for the title figure and much of the story. Nonetheless, this paper could be a unique place to encourage better practice in the literature. Therefore, I would love to see results from tuning on a separate validation set featured more prominently in the paper than in the appendix. The finding that the test set metrics drop significantly when not tuning on the test set is expected, but also a significant insight on its own, as it suggests that robustness generalization results in the literature are inflated. Is it feasible for you to obtain full results on a separate validation set for the camera-ready?

---

> > > ### Author Response · Authors · 2026-04-06
> > >
> > > We sincerely thank the reviewer for the thoughtful acknowledgment of our response and are pleased that all concerns have been fully resolved.
> > >
> > > We are especially grateful for the suggestion to include results on a validation split, since “this paper could be a unique place to encourage better practice in the literature”. We fully agree that tuning on the test set is a highly questionable practice, and we believe our submission can serve as a strong reference point for improved evaluation protocols that are closer to the true performance of certified training going forward.
> > >
> > > We are currently running the main paper experiments tuning on a separate validation set and expect them to end in 3 weeks, which will be before the paper notification. Therefore, we are confident that these results will be ready for the camera-ready deadline. These experiments will allow us to highlight the issue of inflated performance reporting in certified training when improper tuning practices are used. We will summarise these findings in the main paper, with full results provided in the appendix.
> > >
> > > We thank the reviewer again for this valuable suggestion and encouragement, and we are confident that incorporating it will further strengthen the paper and its impact on future work.

---

### Official Review · Reviewer_fVzW · 2026-03-11

**Soundness:** 2
**Presentation:** 2
**Significance:** 2
**Originality:** 2
**Overall Recommendation:** 4
**Confidence:** 2

**Summary:**

This paper argues that current evaluation practices for certified training methods are flawed because they rely on single hyperparameter configurations. The authors propose evaluating methods using Pareto frontiers that capture the full trade-off between natural and certified accuracy. They perform automated multi-objective hyperparameter optimization to generate these frontiers and apply the framework to several existing certified training methods. Their results reveal that many methods were previously under-tuned, and that improvements reported in the literature are less significant than believed. The study provides the first standardized multi-objective comparison of certified training approaches and recommends adopting Pareto-based evaluation as the new standard.

**Compliance With Llm Reviewing Policy:**

Affirmed.

**Final Justification:**

I am satisfied with the authors' response.

**Key Questions For Authors:**

Please see my comments above on the weakness of this work. I expect the authors to respond to those 4 concerns accordingly.

**Limitations:**

See the weakness of this work.

**Strengths And Weaknesses:**

Strength:
1- The paper questions the standard evaluation practice used in certified robustness research. Most prior papers compare methods using a single hyperparameter configuration, which can be misleading. The authors show that this approach can produce incorrect conclusions about which method is better.
2- Instead of reporting one pair (natural/certified) accuracy, the paper proposes evaluating methods using a Pareto frontier. This allows researchers to understand the full trade-off curve between these two accuracies.

Weakness:
1- The paper does not provide a theoretical analysis explaining why certain methods dominate specific regions of the Pareto frontier. They did not comment on why such a trade-off exists.
2- Experiments are mainly conducted on standard small benchmarks such as CIFAR-10. The study does not examine large-scale datasets, transformers, etc. Therefore, scalability is unclear.
3- The work primarily studies IBP-based certified training. How about other certification methods? I have some doubts about the generalization of this methodology.
4- Computing Pareto frontiers requires training many models and extensive hyperparameter optimization. This may be impractical for many research groups with limited resources.

---

> ### Author Rebuttal · Authors · 2026-03-31
>
> We sincerely thank the reviewer for the careful evaluation. We are particularly encouraged that the reviewer recognises that the paper “questions the standard evaluation practice used in certified robustness research” and highlights that comparing methods using a single hyperparameter configuration “can be misleading.” We also appreciate that the reviewer acknowledges the benefit of evaluating methods using a Pareto front, which allows understanding “the full trade-off curve between these two accuracies.” Below, we address the specific concerns raised.
>
> **Q: Lack of theoretical analysis explaining why certain methods dominate specific Pareto regions or why the trade-off exists.** While our work is primarily methodological and empirical, the robustness–accuracy trade-off is a well-established phenomenon in adversarial robustness. Enforcing robustness constraints encourages models to ignore non-robust but predictive features, producing smoother decision boundaries that can reduce performance on clean inputs. For example, Tsipras et al. \[1] show that adversarial robustness can conflict with accuracy, and Zhang et al. \[2] provide a principled characterisation of the trade-off via simultaneous minimisation of natural and robust errors. Ilyas et al. \[3] highlight that robust training suppresses non-robust features useful for clean accuracy. IBP-based methods amplify this effect through over-regularisation: conservative bounds that improve certificates constrain model expressiveness, increasing the observed trade-off. Our contribution is not to resolve these causes but to reveal trade-offs systematically across methods and hyperparameters. By visualising the full Pareto front, our framework lays the groundwork for future theoretical and algorithmic investigations.
>
> **Q: Experiments focus on standard benchmarks (e.g., CIFAR-10); scalability to larger datasets or architectures is unclear.** We agree that evaluating larger datasets and architectures is important. Beyond CIFAR-10, our experiments include TinyImageNet (Fig. 2c) and wider and deeper network variants (Appendix D.2). We will clarify this in the camera-ready version of our paper. Interestingly, simply scaling network size does not improve trade-offs due to accumulated over-approximation errors in IBP \[4], highlighting scalability challenges. CIFAR-10 and similar benchmarks remain standard in certified training, and prior work largely reports results in this regime. Our goal was to enable fair, systematic comparisons in a widely used setting. The proposed Pareto-based evaluation and multi-objective optimisation procedure are not dataset- or architecture-specific and can be applied to larger-scale settings.
>
> **Q: Generalisation to other certified training methods.** Our empirical study focuses on IBP due to its state-of-the-art status for $\ell\_\infty$ guarantees and its many hyperparameters that strongly influence the robustness–accuracy trade-off. This highlights the need for standardised, performant tuning procedures. However, the evaluation framework itself is not specific to IBP: any certified method with tunable hyperparameters and measurable, conflicting objectives can be evaluated via Pareto fronts. This includes Lipschitz-constrained or randomised smoothing methods, where under-approximations or reduced sampling are used. We will explicitly clarify this broader applicability in the camera-ready version and hope the work inspires similar analyses in other certification subfields.
>
> **Q: Computational costs.** Computing Pareto fronts requires multiple models, but the number of evaluations aligns with standard hyperparameter tuning budgets \[5]. Future work can leverage our results without repeating the full evaluation. We also show that good approximations of the Pareto front can be obtained with substantially reduced budgets (Appendix F); for instance, Pareto fronts are recovered with reduced verification timeouts, and reasonable approximations appear after 50% of the optimisation budget.
>
> With this, we hope to have addressed the remaining concerns of the reviewer. By revealing previously hidden performance regimes, enabling fair comparisons, and providing a framework applicable across architectures, datasets, and certification methods, we believe this work offers a valuable foundation for future research in certified robustness.
>
> \[1] Tsipras et al., Robustness May be at Odds with Accuracy, ICLR 2019
>
> \[2] Zhang et al., Theoretically Principled Trade-Off between, ICML 2019
>
> \[3] Ilyas et al., Adversarial Examples are Features not Bugs, NeurIPS 2019
>
> \[4] Mao et al., Understanding Certified Training with Interval Bound Propagation, ICLR 2024
>
> \[5] Mao et al., CTbench: A Library and Benchmark for Certified Training, ICML 2025

---

> > ### Author Rebuttal · Reviewer_fVzW · 2026-04-03
> >
> > Some of my concerns have been resolved. However, the scalability issue still remains. I will increase my score accordingly.

---

> > > ### Author Response · Authors · 2026-04-06
> > >
> > > We thank the reviewer for their thoughtful response and are pleased that we were able to address most concerns. We are particularly grateful for the revised score.
> > >
> > > We agree that scalability is an important point and will sharpen its discussion in the camera-ready. Here, we briefly reiterate why we believe it is less limiting than it may appear.
> > >
> > > Our experiments target the regime where IBP-based certified training is currently practical and widely studied, i.e., datasets up to TinyImageNet with CNN-style architectures, consistent with prior empirical work. At present, certified training and complete verification do not reliably scale to modern architectures such as transformers or large ResNets. Our evaluation therefore focuses on the frontier where these methods remain tractable.
> > >
> > > Importantly, the core components of our framework are agnostic to datasets, architectures, and verification methods. As certified training and verification techniques scale, the same protocol can be instantiated with larger models, different search spaces, improved proxy objectives (e.g., stronger incomplete verifiers), and more advanced HPO strategies. Our primary contribution is methodological: defining a principled way to compare certified training approaches via Pareto fronts and multi-objective HPO under controlled tuning budgets.
> > >
> > > We also explicitly study how to reduce computational cost in the current setting. As shown in Appendix F, (i) reducing complete-verification timeouts by an order of magnitude yields nearly identical Pareto fronts, and (ii) good approximations are typically obtained with roughly half the HPO budget. This suggests that, for larger models, practitioners can adopt more aggressive approximations, e.g., shorter timeouts, fewer trials, or cheaper verifiers, while still avoiding the limitations of single-point tuning. Moreover, to contextualise new methods, researchers only need to run the protocol on their approach and can rely on our results for comparisons to prior work.
> > >
> > > We thank the reviewer again for the constructive feedback and hope this clarifies and mitigates the scalability concerns.

---

### Official Review · Reviewer_vApP · 2026-03-12

**Soundness:** 4
**Presentation:** 2
**Significance:** 4
**Originality:** 3
**Overall Recommendation:** 5
**Confidence:** 4

**Summary:**

This paper provides a framework for pareto-based evaluation of the trade-off between natural and robust accuracy for IBP-based certification methods. In addition to identifying inconsistencies in how many state of the art certified training methods are evaluated, this paper combines many existing approaches into a set of highly pragmatic assumptions about adversarial training, hyperparameters, and varying levels of formal certificates which can be differentially employed for computational efficiency. The result presented is a full-package evaluation applied to several state-of-the-art methods revealing under-evaluation in several key results and providing a highly standard evaluation format to move forward.

**Compliance With Llm Reviewing Policy:**

Affirmed.

**Final Justification:**

I am satisfied with the author's comments, although this does not change my original favorable evaluation. Ultimately I believe that this is a nice study of a commonly confused performance metric and that it is timely, although the narrow focus of the evaluation (both of model and datasets) prevents me from offering a higher score.

**Key Questions For Authors:**

Limiting the epsilon radius of the PGD attack opens up a question
about that as a hyperparameter and alternatives such as Line 247 does
not give the relationship between the PGD radius and the certified
radius -- which I would expect to see. What is the relationship
between the adversarial radius and the certified radius and would it
make more sense to make the adversarial training unlimited and the
radius a deterministic hyperparameter?

In a more broad context, this geometric trade-off potentially
represents a limitation of these results more broadly as the
optimization assumptions made within this benchmark represent a useful
default assumption given our present absence of broader
characterization of useful model features. Adi Shamir's dimpled
manifold hypothesis and Andrew Ilyas' Adversarial Examples are Not
Bugs, They are Features both illustrate a subtle structure of features
which can be used by neural network models. It is incorrect to assume
that neural networks cannot be trained to react to features in diverse
and subtle combinations. This would induce complex geometry that is
not sufficiently accounted for by PGD or IBP based radii. Indeed, more
subtle connections will almost certainly play a key role in the next
generation of verification tools and it is essential that our
benchmark evaluation systems do not depend too strongly on pragmatic
but ultimately false geometric assumptions.

In other words, If we agree that this is a correct evaluation
framework for the flawed SOTA today on limited models up to
tinyimagenet, what will an appropriate evaluation framework look like
for systems that address modern models at scale?

cnn7 is notable in an older result of Muller et al. for over-emphasizing
certified accuracy compared to natural accuracy. This may deserve
discussion rather than reference.

Many works in this space are evaluated on small benchmarks such as
MNIST and CIFAR-10 which lack complexity representative of more modern
problems. How should dataset complexity be incorporated into this
pareto-based evaluation?

**Limitations:**

A few key limitations of this work: Naive geometric assumptions, dataset
size and characterization, and dependence on pragmatic hyperparameter
assumptions, have consequences which could be discussed at greater length.

Computational cost as a limitation is nicely addressed, noting that
this method, by making pragmatic assumptions and algorithmic choices,
uses similar evaluation compute budgets to existing state-of-the art
evaluations.

There appear to be no broad societal negative impacts, although as
stated previously, the research community should still be encouraged
to think broadly about how the benchmark presented differs from the
benchmark that will be ultimately necessary.

**Strengths And Weaknesses:**

Soundness: This paper is well motivated both in its high level focus
on a useful research question and on its pragmatic approach to
implementing and evaluating state of the art certified training
methods. The surrounding literature is thoroughly covered and the
motivation, methods, and background nicely support the empirical
results.

A small point, in Column 1 line 242, where PGD step limitations are
discussed, it would be useful to set up the relationship between
adversarial attack epsilon and certification radii as both being
determined by the same inherent geometry of model classification.

More broadly, this is a limitation on significance where ultimately
radii implicitly assume a regular and general geometric structure
which possibly will not remain true as models get more nuanced to
address adversarial problems. This limitation deserves careful
consideration in the discussion throughout the paper.

Presentation:In general, the presentation is excellent with solid
prose and discussion that focuses with correct emphasis on key
aspects. In addition, the literature presentation is solid and nicely
sets up scope.

A small nit-pick: It can be argued that the state of the art are
hybrid methods that are approaching full Branch and Bound, TAPS, or
using actively optimized bounds like Partial Crown.

[1] Eiras, F., Bibi, A., Bunel, R., Dvijotham, K. D., Torr, P., & Kumar, M. P. (2023). Efficient error certification for physics-informed neural networks. arXiv preprint arXiv:2305.10157.

Ultimately these methods are focused on different areas of the SOTA,
particularly in physics informed machine-learning, however since this
is a key application of certified training, the scope of the claims in
this paper should be more carefully differentiated.


Figure 3 needs major adjustment to its formatting. For colorblind
readers, (d) (h) and (l) have extremely low
brightness contrast and are very difficult to discern. The lack of legends
for the other plots is not helpful. It would be STRONGLY preferable to
have 3 rows with larger figures on each row. In addition, the
different color-pairs per plot seem more confusing than useful.
Finally, it is difficult from what is presented, to understand the
combined influence of optimization choices (differing PGD algorithms
and constraints on IBP) versus under-explored hyperparameter
regions. It would be preferable to isolate for this by adding
intermediate points showing identical hyperparameters to the
literature and only swapping out the optimization method. Then the
rest of the mapped frontier should be shown.

Significance: Given the scope of available literature in this space
and the complex arbitrary choices often made during evaluation, this
paper is both timely and significant. This work has the potential to
save significant time across the community.

Originality: This work appears to be a novel combination of evaluation
approach, pragmatic and effective algorithmic choices, and benchmark
evaluations.

---

> ### Author Rebuttal · Authors · 2026-03-31
>
> We sincerely thank the reviewer for the thoughtful and detailed evaluation. We are particularly encouraged that the reviewer considers the paper “well motivated”, highlighting both the “useful research question” and the “pragmatic approach to implementing and evaluating state-of-the-art certified training methods.” We also appreciate the recognition that the paper provides a “highly standard evaluation format to move forward” and that the work is “timely and significant” with the potential to “save significant time across the community.” Below we address the reviewer’s comments and questions.
>
> **Q: Geometric assumptions underlying radius-based robustness evaluations.**
>
> The reviewer raises an important broader point that radius-based certification implicitly assumes a particular geometric structure of the decision boundary, and that future models may exhibit more complex feature geometries. We agree that this is an important conceptual limitation of current certification methods. Our work does not attempt to resolve these underlying assumptions, but rather provides a framework for systematically evaluating certified training approaches agnostic to the underlying threat models or certification approaches. In either case, our proposed shift to Pareto-based evaluations remains crucial for assessing techniques across the whole trade-off space. We will discuss the limitations regarding the simple geometries of the threat models employed in the conducted experiments in more detail.
>
> **Q: Applicability of the framework to future large-scale models and systems.**
>
> As the reviewer notes, the present study evaluates “limited models up to TinyImageNet.” Our goal was to establish a standardised evaluation methodology in the regime where certified training methods are currently most mature and widely studied. Importantly, the Pareto-based evaluation framework itself is independent of model architecture, dataset size, or verification method. As certification tools scale to larger models and more complex properties, the same evaluation principle, i.e., characterising performance across the full trade-off between natural and certified accuracy, remains directly applicable. We will clarify this broader perspective in the discussion.
>
> **Q: Fig. 3 formatting and visualisation clarity.**
>
> We thank the reviewer for the detailed suggestions. We agree that the readability of Fig. 3 can be improved, particularly regarding brightness contrast for colourblind readers and the absence of legends, which we will address for the camera-ready version.
>
> **Q: Relationship between adversarial attack radius (PGD ε) and certification radius.**
>
> We thank the reviewer for pointing out that the relationship between these quantities could be clarified. In our experiments, the adversarial attack radius and the certified radius correspond to the same perturbation budget ε under the chosen threat model, but the radius of the employed PGD attack can be scaled to yield bigger loss values and, thereby, shifting higher focus on PGD bounds in SABR and MTL-IBP.
>
> **Q: CNN7 architecture and prior observations.**
>
> We thank the reviewer for pointing out the observations by Mao et al. \[1] regarding the CNN7 architecture. We will incorporate a short discussion clarifying this point in the revised manuscript.
>
> We thank the reviewer again for the constructive feedback and positive assessment. We believe that the suggested clarifications and presentation improvements will further strengthen the manuscript and help the community adopt more systematic evaluation practices for certified training methods, including those applicable to geometrically more complex threat models.
>
> \[1] Mao et al., Understanding Certified Training with Interval Bound Propagation, ICLR 2024

---

> > ### Author Rebuttal · Reviewer_vApP · 2026-04-04
> >
> > This has addressed most of the primary questions from initial review.
> >
> > With emphasis on applicability of this work to further models and developments, I would like to see more discussion about the precise limitations and considerations for researchers considering applying this method to larger and more complicated models. This should combine the CNN7 discussion, relationship between adversarial attack radius, and certified radius, and focus on how exploration of the pareto front may change under different choices.

---

> > > ### Author Response · Authors · 2026-04-06
> > >
> > > We thank the reviewer for their thoughtful response and are encouraged that the main concerns were largely resolved in our rebuttal.
> > >
> > > We agree that a clearer discussion of the limitations and assumptions of our empirical evaluation would strengthen the paper, and we will emphasise that the proposed evaluation scheme itself is the key contribution of our paper beyond the specific experiments conducted on the state of the art in certified training.
> > > First, we will clarify the role of CNN7 (and its variants in the appendix). We use this architecture because it is the de facto standard in prior work, enabling fair comparison. At the same time, we will explicitly note that CNN7 is substantially less complex than modern models and overemphasises certified accuracy relative to natural accuracy, consistent with prior findings and our own architectural sensitivity study. We will further highlight that, for larger models, both the shape of the Pareto front (e.g., reduced certified accuracy) and the computational budget for hyperparameter optimisation are expected to change. This motivates our framework as a general template, combining multi-objective HPO, proxy verification, and constrained search, rather than a fixed recipe.
> > >
> > > Second, we will clarify the relationship between certified and adversarial radius. The certified radius is fixed by the local robustness specification, while the adversarial radius is treated as a tunable parameter (via a scaling factor) to control the influence of PGD and IBP bounds. Our evaluation protocol is agnostic to the specific search space; in principle, adversarial training could be made unbounded and the radius fixed deterministically. However, we adopt hyperparameter search spaces consistent with the originally proposed methods, noting that alternative choices may yield different, not directly comparable results.
> > >
> > > Finally, we will expand the discussion of how Pareto-front exploration may change under more complex or less geometric threat models. Our current study focuses on $\ell_\infty$ local robustness with standard geometric assumptions, as common in IBP-based certification. For richer settings, e.g., involving more complex decision-boundary geometry or feature-based views of adversarial examples, the definition and dimensionality of the Pareto front may need to evolve, potentially incorporating robustness to semantic or distributional shifts or other domain-specific guarantees. Nevertheless, the central idea of evaluating methods via their full trade-off surfaces, rather than single operating points, remains unchanged.
> > >
> > > Due to space constraints, we will briefly address these points in the main text and provide a more detailed discussion in the appendix. We thank the reviewer again for these valuable suggestions and are confident they will improve the paper.

---

### Decision · Program_Chairs · 2026-04-30

**Decision:**

Accept (regular)

**Comment:**

This paper’s main contribution is an evaluation framework for IBP-based certified training that replaces single-point reporting with Pareto-front analysis over the natural-versus-certified accuracy trade-off. Instead of judging methods at one arbitrarily chosen hyperparameter setting, it evaluates the full trade-off surface. The paper shows that previously reported configurations are often under-tuned and reveals new performance regimes.

The main weakness raised by the reviewers includes the relatively small datasets, the lack of theoretical analysis, and the cost of Pareto frontier evaluation. The AC believes that these datasets are commonly used in prior work on certified defense and are not a major weakness (due to the cost of the experiments), and it is not clear what theoretical analysis would be directly beneficial due to the empirical nature of the work. Although the cost of Pareto-front analysis is indeed high, the AC believes the value of the study is to actually show this trade-off for a wide range of methods and settings, and discover useful findings and inspire future research. The authors have also demonstrated that approximations can be obtained with just a fraction of the search budget.

The authors have thoroughly and adequately addressed remaining concerns during the rebuttal, and there are no outstanding fatal flaws. I therefore recommend accepting this paper. For the camera-ready version, the authors should be sure to discuss validation-set tuning results and clarify more explicitly the limitations and computational demands of the proposed approach.